EMBO
*reports*

# H4K20me1 and H3K27me3 are concurrently loaded onto the inactive X chromosome but dispensable for inducing gene silencing

Sjoerd J D Tjalsma[1,†], Mayako Hori[2,†], Yuko Sato[2,3,†] (iD), Aurelie Bousard[1], Akito Ohi[2], Ana Cláudia Raposo[4], Julia Roensch[1], Agnes Le Saux[1], Jumpei Nogami[5], Kazumitsu Maehara[5] (iD), Tomoya Kujirai[6], Tetsuya Handa[3] (iD), Sandra Bagés-Arnal[7], Yasuyuki Ohkawa[5] (iD), Hitoshi Kurumizaka[6], Simão Teixeira da Rocha[4] (iD), Jan J Żylicz[1,7,8,*] (iD), Hiroshi Kimura[2,3,**] (iD) & Edith Heard[9,10,***] (iD)

## Abstract

During X chromosome inactivation (XCI), in female placental mammals, gene silencing is initiated by the *Xist* long non-coding RNA. *Xist* accumulation at the X leads to enrichment of specific chromatin marks, including PRC2-dependent H3K27me3 and SETD8-dependent H4K20me1. However, the dynamics of this process in relation to *Xist* RNA accumulation remains unknown as is the involvement of H4K20me1 in initiating gene silencing. To follow XCI dynamics in living cells, we developed a genetically encoded, H3K27me3-specific intracellular antibody or H3K27me3-mintbody. By combining live-cell imaging of H3K27me3, H4K20me1, the X chromosome and *Xist* RNA, with ChIP-seq analysis we uncover concurrent accumulation of both marks during XCI, albeit with distinct genomic distributions. Furthermore, using a *Xist* B and C repeat mutant, which still shows gene silencing on the X but not H3K27me3 deposition, we also find a complete lack of H4K20me1 enrichment. This demonstrates that H4K20me1 is dispensable for the initiation of gene silencing, although it may have a role in the chromatin compaction that characterises facultative heterochromatin.

**Keywords** embryonic stem cells; H4K20me1; heterochromatin; polycomb; X inactivation
**Subject Category** Chromatin, Transcription & Genomics

## Introduction

Dynamic changes to the chromatin landscape allow for timely execution of developmental and differentiation programmes. Indeed, chromatin modifiers often reinforce signalling cues to initiate and/or maintain an exact transcriptional outcome (Stasevich *et al*, 2014; Zylicz *et al*, 2015; Jambhekar *et al*, 2019). One powerful model, where such regulation takes place, is X chromosome inactivation (XCI) in female mammals. Although much is known about the involvement of certain chromatin marks in initiating XCI (reviewed in Zylicz & Heard, 2020), it was thus far impossible to track chromatin rearrangements and the inactive X chromosome (Xi) in living cells. What is more, the role and dynamics of some histone modifications during XCI remains enigmatic.

In eutherian female mammals, XCI is initiated at around the time of implantation, when each cell randomly inactivates one of the two X chromosomes (Lyon, 1962). This process can be modelled *in vitro* by differentiating female mouse embryonic stem cells (ESC) (Rastan & Robertson, 1985). Both *in vivo* and *in vitro*, XCI depends on the coating of the Xi by a long non-coding RNA called *Xist* (X-inactive-specific transcript) (Penny *et al*, 1996). *Xist* RNA accumulation along the Xi induces not only the silencing of over a 1,000 genes but also a cascade of chromatin alterations (reviewed in Zylicz & Heard,

1 Mammalian Developmental Epigenetics Group, Institut Curie, CNRS UMR3215, INSERM U934, PSL University, Paris, France
2 Graduate School of Bioscience and Biotechnology, Tokyo Institute of Technology, Yokohama, Japan
3 Cell Biology Center, Institute of Innovative Research, Tokyo Institute of Technology, Yokohama, Japan
4 Faculdade de Medicina, Instituto de Medicina Molecular, João Lobo Antunes, Universidade de Lisboa, Lisboa, Portugal
5 Division of Transcriptomics, Medical Institute of Bioregulation, Kyushu University, Fukuoka, Japan
6 Institute for Quantitative Biosciences, The University of Tokyo, Tokyo, Japan
7 The Novo Nordisk Foundation Center for Stem Cell Biology, Copenhagen, Denmark
8 Department of Physiology, Development and Neuroscience, University of Cambridge, Cambridge, UK
9 EMBL Heidelberg, Heidelberg, Germany
10 Collège de France, Paris, France
 *Corresponding author. Tel: +45 23839889; E-mail: jan.zylicz@sund.ku.dk
 **Corresponding author. Tel: +81 45 924 5742; E-mail: hkimura@bio.titech.ac.jp
 ***Corresponding author. Tel: +49 6221 3878 201; E-mail: edith.heard@embl.org
 †These authors contributed equally to this work

2020). *Xist* is a modular non-coding RNA with specific regions playing distinct roles. Its 5′ A-repeat region is vital for the induction of gene silencing (Wutz *et al*, 2002). It does so by recruiting SPEN, a key RNA-binding protein, which integrates many repressive complexes including NCOR/SMRT and HDAC3 (McHugh *et al*, 2015; Zylicz *et al*, 2019; Dossin *et al*, 2020). On the other hand, the B and to a lesser extent C repeat regions are vital for the recruitment of Polycomb-group repressive complexes (PRC) (Bousard *et al*, 2019; Colognori *et al*, 2019; Nesterova *et al*, 2019). Indeed, upon *Xist* RNA coating Xi becomes rapidly enriched with the PRC1-dependent H2AK119Ub and only subsequently with PRC2-dependent H3K27me3 (Plath *et al*, 2003; Silva *et al*, 2003; de Napoles *et al*, 2004; Zylicz *et al*, 2019). Recent data indicate that the B and C repeat region of *Xist* RNA directly recruits hnRNPK, which in turn binds non-canonical PRC1 allowing for rapid H2AK119Ub deposition (Almeida *et al*, 2017; Pintacuda *et al*, 2017; Bousard *et al*, 2019; Colognori *et al*, 2019). H2AK119Ub could potentially be recognised by PRC2 cofactors allowing for *de novo* deposition of H3K27me3 (Blackledge *et al*, 2014). Importantly, Polycomb accumulation is dispensable for the initiation of gene silencing, as *Xist* mutants lacking the B and C repeat region can still induce XCI albeit with slightly lower efficiency (Bousard *et al*, 2019; Nesterova *et al*, 2019). Instead, PRC2 enables stable maintenance of the repressed state, particularly in the context of extra-embryonic lineages (Kalantry *et al*, 2006). The dynamics of accumulation of PcG marks have been studied using ChIP-seq, although it remains unclear how precisely this correlates with *Xist* RNA coating of the Xi. Indeed, the time that lapses between *Xist* RNA upregulation and H3K27me3 deposition is not known. Nor how this relates to other repressive histone marks.

Another modification that rapidly accumulates on the Xi is H4K20me1 (Kohlmaier *et al*, 2004), although its role in XCI is not well understood. H4K20me1 is deposited by SETD8 (also called Pr-Set7) (Nishioka *et al*, 2002) and is thought to regulate a variety of processes relating to transcription, chromosome condensation, DNA replication and the DNA damage response (reviewed in van Nuland & Gozani, 2016). Consistent with the involvement of H4K20me1 in cell division, it transiently accumulates during G2 and mitosis, whereas H4K20me2 and -me3 do not show similar fluctuations during the cell cycle (Wu *et al*, 2010). This dynamics is achieved by stable expression of SETD8 during G2/M when its phosphorylation prevents the interaction with the anaphase-promoting complex (APC). During late mitosis, CDC14 dephosphorylates SETD8 thus stimulating its degradation by APC. On the other hand, PHF8, H4K20me1 demethylase, is degraded by APC during early G2 (Lim *et al*, 2013) together allowing for efficient deposition of H4K20me1 and successful progression through mitosis. Furthermore, H4K20me1 accumulates at centromeres where it promotes kinetochore assembly and thus correct chromosome segregation (Hori *et al*, 2014). The regulation of H4K20me1 levels could also be achieved by modulating the efficiency of further methylation by SUV420H1/2 or demethylating H4K20me2/3 by specific enzymes. DPY-21 mediates the latter a reaction in *Caenorhabditis elegans*, and its two mouse orthologs (RSBN1/RSBN1L) retain specific demethylase activity *in vitro* (Brejc *et al*, 2017). However, the biological function of these enzymes remains unknown. The study of H4K20me1 during XCI has been hampered by the fact that SETD8 is absolutely required for the progression through mitosis and embryos lacking it die before the onset of XCI (Oda *et al*, 2009; Shikata *et al*, 2020).

While the genomic distribution of H4K20me1 at a stably inactivated X was previously described (Calabrese *et al*, 2012), the dynamics of H4K20me1 accumulation during XCI initiation in relation to *Xist* RNA and H3K27me3 deposition remains unclear. Furthermore, the precise distribution of first H4K20me1 enrichment along the X and the molecular mechanisms underlying its dynamic accumulation on the Xi have not previously been investigated.

To address these questions, it is important to follow the spatio-temporal dynamics of histone modifications as they accumulate on the X chromosome in living cells. To this end, we have previously developed genetically encoded, modification-specific intracellular antibodies or mintbodies, by fusing a single-chain variable fragment (scFv) of the specific antibody with a fluorescent protein (Sato *et al*, 2013; Sato *et al*, 2016). The expression of mintbodies enables tracking of changes in endogenous histone modification levels without affecting cell cycle progression and developmental processes in such model organisms as fission yeast, nematode, drosophila and zebrafish (Sato *et al*, 2013; Arai *et al*, 2017). Here, to follow XCI dynamics in living cells, we developed a mintbody specific for the PRC2-dependent H3K27me3. In addition, we implemented sgRNA-dCas9 system to visualise both X chromosomes in living cells (Chen *et al*, 2013; Anton *et al*, 2014). We used these novel tools in combination with an H4K20me1-specific mintbody (Sato *et al*, 2016; Sato *et al*, 2018) and a method to visualise endogenous *Xist* RNA in live cells (Masui *et al*, 2018; Dossin *et al*, 2020). This allowed us to show that there are striking similarities in the accumulation dynamics of both H3K27me3 and H4K20me1 marks on the Xi. Further ChIP-seq analysis confirmed concurrent accumulation of H4K20me1 and H3K27me3 during XCI albeit with rather distinct genomic distributions and correlations with gene silencing dynamics. By using cells expressing mutant *Xist* RNA, we demonstrate that H4K20me1 accumulation, just like that of PcG-dependent H3K27me3, relies on the *Xist* RNA B and C repeats. This also reveals that both marks are dispensable for the initiation of gene silencing. Taken together, our analysis uncovers that the H4K20me1 and H3K27me3 histone marks accumulate at the Xi with comparable dynamics but with rather different distributions. Reliance of both marks on the *Xist-BC* region for their enrichment suggests a mechanistic link between Polycomb and H4K20me1 accumulation during facultative heterochromatin formation on the X chromosome. These observations also imply that the general function of H4K20me1 may be in chromatin compaction that characterises facultative heterochromatin rather than in the initiation of gene silencing.

## Results

### H3K27me3 mintbody (2E12LI) specifically tracks H3K27me3 in living cells

Methods for tracking polycomb-dependent histone marks in living cells have been lacking up until now. Such tools would be hugely beneficial for the study of dynamic epigenetic processes such as XCI. To address this, we decided to generate an H3K27me3-specific mintbody. Upon screening of mouse hybridomas, we selected a 2E12 clone, which expresses an H3K27me3-specific antibody. We determined the cDNA sequence of the IgG heavy and light chains in 2E12 by deep sequencing (Kuniyoshi *et al*, 2016) and cloned the variable fragments by PCR to construct a mintbody expression vector (Figs 1A

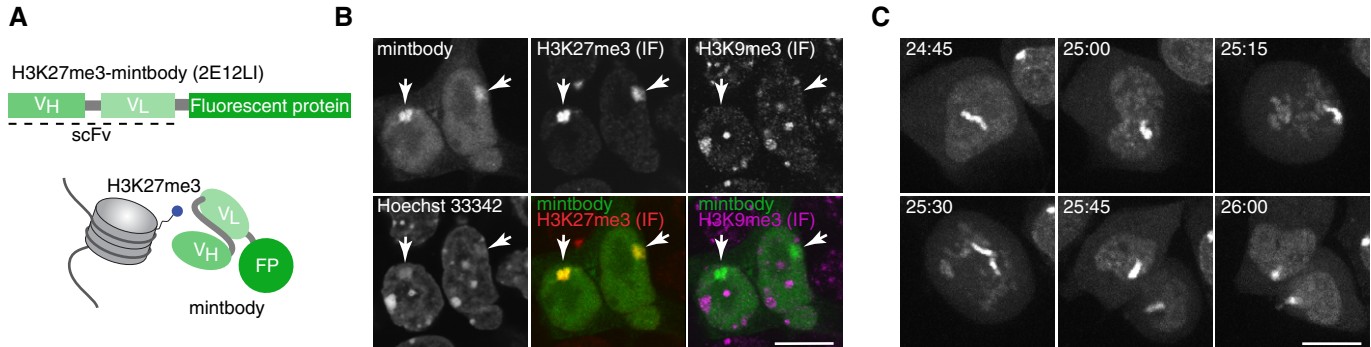

**Figure 1. Establishing H3K27me3 mintbody to visualise H3K27me3 in living cells.**

A   Schematic representation of the mintbody. H3K27me3-specific single-chain variable fragment (scFv) is genetically fused with a fluorescent protein (FP).

B   Immunofluorescence (IF) validation of mintbody specificity. Mouse MC12 cells, which stably express H3K27me3-mintbody (sfGFP), are labelled with antibodies specific for H3K27me3 (Cy5) and H3K9me3 (Cy3). DNA is stained with Hoechst33342. Single confocal sections are shown. Arrows mark Xi. Scale bar = 10 μm.

C   Time-lapse imaging of a dividing MC12 cell stably expressing H3K27me3-mintbody (sfGFP). Projection images of 7 confocal sections with 2 μm intervals are shown with elapsed time (hh:mm). Scale bar = 10 μm.

and EV1A). To validate that the mintbody properly localises to H3K27me3-enriched chromatin in living cells, we used mouse terato-carcinoma cell line MC12 (Abe *et al*, 1988). MC12 is a mixture of diploid and tetraploid female cells harbouring one or two Xi, on which the mintbody is expected to be concentrated. When expressed in MC12, the superfolder (sf) GFP version of 2E12 mintbody (2E12-sfGFP) localised mostly in cytoplasm and tended to form aggregates (Fig EV1B, left), suggesting that 2E12-sfGFP was not able to function in cells, probably due to a typical problem in folding and/or structural stability associated with the intracellular expression of scFv (Cattaneo & Biocca, 1999; Ewert *et al*, 2004). To prevent aggregate formation, we performed PCR-based random mutagenesis and transient transfection into MC12 cells. One mutant that localised preferentially in the nucleus with focal enrichment (presumably Xi) had Met 86 to Leu substitution in the heavy chain (M86L) (Fig EV1B, middle). Changes in localisation suggested that M86L mutation improved the mintbody function to bind to H3K27me3. To gain insight into the contribution of M86L mutation in scFv structure, we looked into the atomic models built based on the X-ray structure of the most similar scFv (Fig EV1C, left). The modelling indicated that Met 86 is located within a hydrophobic core of the heavy chain, and its substitution to Leu appears to fill the space in the core better than the original Met to strengthen hydrophobic interactions. We have previously observed that the hydrophobicity of an amino acid in a hydrophobic core is important for the functionality of a H4K20me1-mintbody (Sato *et al*, 2016). However, the 2E12M86L-sfGFP was still not exclusively located in the nucleus and we were unable to obtain cells stably expressing it. Thus, we decided to further strengthen scFv folding with an additional mutation at Met 158 in the light chain's hydrophobic core (Fig EV1C, right). We constructed three mutants in which Met 158 was substituted to Ile, Leu and Val in 2E12M86L. Among the three mutations, M185I mutant was the most enriched in the nucleus with evident foci (Fig EV1B, right). To confirm that nuclear foci of 2E12M86L, M185I (2E12LI)-sfGFP represent H3K27me3-enriched Xi, we employed immunofluorescence (IF) on transiently expressing cells using previously characterised specific antibodies directed against H3K27me3 and H3K9me3 (CMA327 and CMA318, respectively (Chandra *et al*, 2012)). The nuclear foci were colocalised with

a H3K27me3-specific antibody, but not with H3K9me3-specific antibody that is concentrated in Hoechst-dense pericentromeric heterochromatin (Fig 1B). These data are consistent with the mintbody foci representing Xi in living cells. The utility of 2E12LI mintbody was further demonstrated by time-lapse imaging of stably expressing MC12 cells, which allowed for the tracking of Xi during interphase and mitosis (Fig 1C and Movie EV1).

To further validate the specificity of 2E12LI to H3K27me3, we performed both biochemical and cell-based assays. We first evaluated the binding specificity of bacterially expressed and purified 2E12LI-sfGFP *in vitro* using a modified histone peptide array (Fig EV2A). H3K27me3-containing peptides were highlighted over other peptides, regardless of the neighbouring R26 modifications (Fig EV2B). The 2E12LI-sfGFP binding was, however, occluded by S28 phosphorylation (S28ph), as commonly observed for methyl-specific antibodies (Kimura *et al*, 2008; Hayashi-Takanaka *et al*, 2011; Kimura, 2013). Next, we assessed whether the purified 2E12LI-sfGFP can selectively bind to H3K27me3 in cells by manipulating the level of specific methylation. HeLa cells were transfected with HaloTag-tagged lysine demethylases KDM6B and KDM4D and fixed for staining with the specific antibodies or 2E12LI-sfGFP. Consistently with the substrate specificity of these KDMs, IF indicated that HeLa cells overexpressing Halo-KDM6B and Halo-KDM4D exhibited drastic decrease of H3K27me3 and H3K9me3, respectively (Fig EV2C and D). The result of staining with purified 2E12LI-scFv was similar to H3K27me3-specific antibody, showing decreased levels of H3K27me3 by Halo-KDM6B overexpression and no changes in H3K27me3 by Halo-KDM4D (Fig EV2E and F). Taken together with the immunofluorescence pattern and biochemical analysis, we concluded that 2E12LI-sfGFP selectively binds to H3K27me3 over the other modifications including H3K9me3. For convenience, we now call 2E12LI fused with a fluorescent protein as H3K27me3-mintbody.

## Simultaneous tracking of X chromosome loci, H3K27me3 and H4K20me1 during XCI

In order to visualise the dynamics of histone modifications during XCI, we decided to develop a method to identify the X chromosome

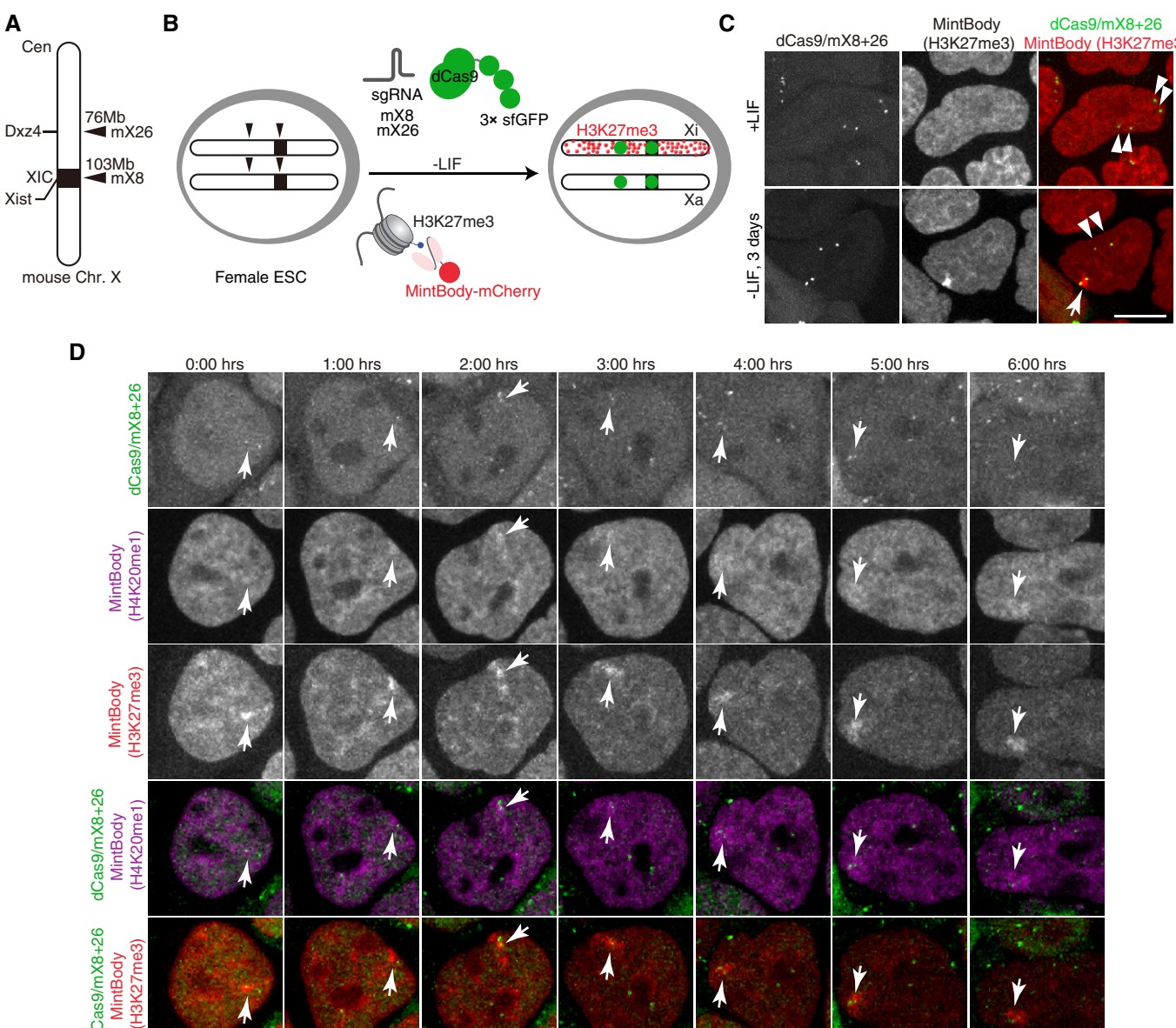

**Figure 2. Simultaneous visualisation of histone marks and X chromosome loci in living cells.**

A   Schematic diagram of CRISPR/dCas9-3×sfGFP targeting loci on mouse X chromosome. sgRNA mX26 and mX8 target microsatellite repeats on *Dxz4* and *Xist* loci, respectively.

B   Experimental design: female ESCs are engineered to stably express two sgRNAs, dCas9-3xsfGFP and a nuclear localisation signal (NLS)-fused H3K27me3-mintbody (mCherry). Upon ESC differentiation by LIF withdrawal H3K27me3 accumulation occurs at the inactivating X (Xi) but not at the active X (Xa). Green foci allow marking of two X-linked loci (*Dxz4* and *Xist*).

C   Live imaging of female mESCs as in (B) in undifferentiated conditions (+LIF) or upon 3 days of LIF withdrawal. Double arrowheads mark Xa and single arrowhead marks Xi. Shown are maximum intensity projections of 11 z-plane confocal sections. Scale bar = 10 μm.

D   ESCs as in (B) were engineered to express H3K27me3-mintbody (SNAP/JF646) and H4K20me1-mintbody (mCherry). Cells were cultured in the absence of LIF for 2 days. Time-lapse images for 12 z-plane confocal stack were acquired every 1 h. Maximum intensity projection images are shown with elapse time (hh:mm). Arrowheads mark Xi. Scale bar = 10 μm.

in live cells. We used an sgRNA-dCas9 (single guide RNA-nuclease dead Cas9) system to label X-linked loci in living cells (Anton *et al*, 2014; Ma *et al*, 2015). We screened repetitive sequences located specifically on the mouse X chromosome and chose 32 microsatellite sequences as target candidates. The expression vectors for

sgRNAs that bind to the sequences were constructed and cotransfected with an expression vector for EGFP-tagged dCas9 into MC12 cells. Among them, 4 sgRNAs (mX2, mX8, mX18 and mX26) enabled visualising X-linked loci in living cells (Fig EV3). We selected mX8 (near *Xist* locus) and mX26 (*Dxz4* locus) to label X

chromosomes and used 3×sfGFP-tagged dCas9 (3×sfGFP-dCas9), to amplify the signal for further analyses using ESCs.

To track H3K27me3 during X inactivation, we established female mouse ESCs (PGK12.1) expressing two sgRNA (mX8 and mX26), 3×sfGFP-dCas9 and H3K27me3-mintbody (mCherry version) (Fig 2 A and B). In undifferentiated ESCs, X chromosome loci were identified as 2 pairs of 3×sfGFP-dCas9 spots and H3K27me3-mintbody was not enriched in either allele (Fig 2C, upper panels). In contrast, 3 days after withdrawal of leukaemia inhibitory factor (LIF), H3K27me3-mintbody accumulated around one pair of 3×sfGFP-dCas9 spots (Fig 2C, lower panels). This result demonstrated that H3K27me3-mintbody together with the sgRNA-dCas9 system allows for tracking the dynamics of histone modification in living and differentiating cells. Since H4K20me1, unlike H4K20me2/3, also becomes enriched on Xi (Fig EV4A; Kohlmaier *et al*, 2004), we sought to reveal the relative accumulation kinetics of the H3K27me3 and H4K20me1 histone modifications during XCI. We established ESCs expressing the two sgRNAs, 3×sfGFP-dCas9, H3K27me3-mintbody (SNAP-Tag version, visualised with JF646) and H4K20me1-mintbody (mCherry version, (Sato *et al*, 2016)) and induced differentiation (3d LIF withdrawal). Time-lapse imaging revealed that whenever H3K27me3 accumulation was visible at the Xi, some level

of H4K20me1 enrichment was also discernible (Fig 2D), suggesting that both repressive marks accumulate concurrently during XCI. This initial analysis in differentiating female ESCs confirmed the successful use of both H3K27me3- and H4K20me1-mintbodies to follow enrichment of these two marks on the X chromosome. However, given the asynchronous nature of random XCI in differentiating ESCs and the rather weak signal from the tagged X-linked loci, it was challenging to distinguish the Xi from the active X (Xa) prior to a significant accumulation of both repressive chromatin marks on the Xi. To ameliorate our analysis, we therefore decided to follow *Xist* RNA itself in living cells alongside H3K27me3 or H4K20me1.

## H3K27me3 accumulates concurrently with H4K20me1 at the Xi

In order to track the relative dynamics of H3K27me3, H4K20me1 and *Xist* RNA during XCI, we combined the use of mintbodies with the inducible *Xist*-Bgl system (Masui *et al*, 2018; Dossin *et al*, 2020; Fig 3A). The latter model is based on the hybrid (*Mus musculus castaneus* x C57BL/6) TX1072 female ESCs line allowing for doxycycline (DOX) induction of the endogenous *Xist* gene from C57BL/6 (*B6*) allele (Schulz *et al*, 2014). By adding DOX, we can induce *Xist*

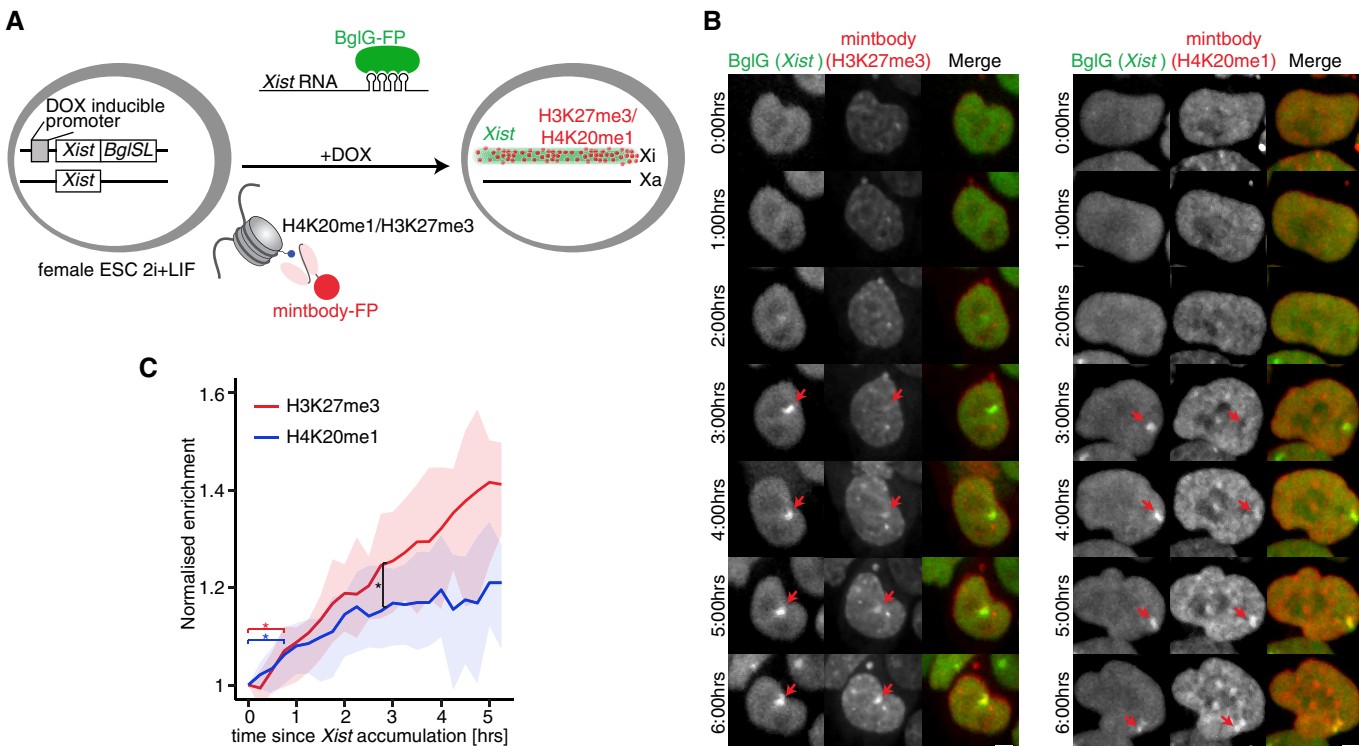

**Figure 3. Simultaneous visualisation of histone marks and *Xist* RNA in living cells.**

A  Schematic representation of the experimental design. Female mouse ESC line was used, in which *Xist* can be induced from one allele (TX1072) and this allele also harbours an array of 18 Bgl stem loops (BglSL) inserted into the 7[th] exon of *Xist* gene. BglG fused to a fluorescent protein (GFP or mCherry) detects *Xist* RNA as it binds to BglSL. Cells also stably express a mintbody allowing the detection of H3K27me3 (GFP) or H4K20me1 (mCherry).

B  Cells were treated with DOX to induce *Xist* expression and time-lapse imaging was performed with images acquired every 15 min. Maximum intensity projection images are shown with elapse time (hh:mm). Arrowheads mark Xi accumulating *Xist* RNA. Scale bar = 5 μm.

C  Live imaging analysis of average H3K27me3 (red) and H4K20me1 (blue) accumulation at the Xi. Average normalised mintbody enrichment is shown with shading representing 25 and 75 quartiles. Signal was calculated starting from the first accumulation of *Xist* RNA. At least 30 cells were analysed. *unpaired *t*-test *P*-value < 0.05.

expression and thus reduce the level of heterogeneity observed during XCI in differentiating ESCs. In addition, 18 Bgl RNA stem loops (BglSL) were knocked into the 7th exon of DOX-inducible *Xist* (Fig EV4B) (Dossin *et al*, 2020). To track *Xist* RNA accumulation, BglSL are visualised by a BglG protein fused to either EGFP or mCherry and expressed from the *Rosa26* or *TIGRE* locus (Fig EV4C and D). The cell lines enabling *Xist* RNA visualisation are named TX-*Xist*-EGFP and TX-*Xist*-mCherry. These cell lines were used to generate stably expressing *PiggyBac* transgenes for the H3K27me3- or H4K20me1-mintbodies (Fig EV4C and D). Two ESC lines, TX-*Xist*-EGFP; H4K20me1-mCherry and TX-*Xist*-mCherry; H3K27me3-sfGFP were used for subsequent analyses. Importantly, the use of *Xist*-BglSL or the mintbodies did not affect the efficiency of gene silencing during XCI (Fig EV4E).

In order to visualise the earliest events following *Xist* RNA accumulation, we started live imaging of ESCs 5 min after DOX induction (Fig 3B, Movies EV2–EV3). Imaging was performed for at least 6 h in 15-min intervals allowing for single-cell tracking and detailed quantitative analysis. *Xist* RNA domains appeared after 2–6 h of DOX induction (Fig EV4F) and were segmented using the *Xist*-EGFP/mCherry signal. The mintbody enrichment within such domains was measured and followed in individual cells throughout the length of the experiment (Fig 3C). Of note, unlike H3K27me3-mintbody, H4K20me1-mintbody shows increased nuclear signal during G2/M phase of the cell cycle, thus tracking the oscillations in H4K20me1 levels (Sato *et al*, 2016). We thus excluded mitotic cells from the analysis due to very high overall levels of H4K20me1. Next, we followed the enrichment of H3K27me3/H4K20me1 signal in at least 30 nuclei individually synchronised to the time point when *Xist* RNA enrichment was first observed (i.e., typically 2–6 h after DOX addition). A significant enrichment of both histone marks was observed within the *Xist* RNA domain about 45 min after *Xist* RNA appearance (adj. *P*-value < 0.05; Fig 3B and C). Initial accumulation of H3K27me3 and H4K20me1 followed very similar dynamics. However, after 2.5 h the two marks significantly diverged (Fig 3C). Indeed, H3K27me3 continued to rapidly accumulate, while the increase in H4K20me1 enrichment significantly slowed down (Fig 3C). Thus, we have successfully performed simultaneous tracking of *Xist* RNA and repressive histone marks. In summary, the enrichment of both H3K27me3 and H4K20me1 is delayed compared to *Xist* RNA accumulation by about 45 min. H3K27me3 shows a continued accumulation over time while that of H4K20me1 eventually slows down. This reveals that the two marks share similar dynamics of enrichment but only at the early stages of XCI.

## Allele-specific native ChIP-seq reveals similarities between H3K27me3 and H4K20me1 accumulation dynamics

Given the above results indicating that although H3K27me3 and H4K20me1 are both initially enriched soon after *Xist* RNA accumulation, their patterns diverge after a few hours, we decided to investigate the molecular distributions of both these marks along the Xi. To this end, we performed allele-specific native ChIP-seq (nChIP-seq) for H4K20me1 in the hybrid female cell line TX1072 (Schulz *et al*, 2014). Thanks to an inducible *Xist* gene on the B6 chromosome, DOX addition leads to rapid gene silencing and chromatin mark alterations that can be measured allelically thanks to the high rate of polymorphism between the B6 and Cast X chromosomes (Zylicz *et al*, 2019). Following

DOX treatment, we tracked H4K20me1 accumulation on the *Xist* RNA-coated B6 allele across five time points at up to 4-h resolution on biological duplicates (Fig 4A). We compared all our results with a matched published dataset of H3K27me3 (Zylicz *et al*, 2019). Importantly, we validated the specificity of H4K20me1 antibody using a peptide array (Fig EV5A) and controlled for *Xist* RNA induction efficiency (Fig EV5B). Upon sequencing, reads were split according to content of allele-specific single nucleotide polymorphisms (SNPs: *B6* mapping to Xi; *Cast* mapping to Xa). Such allelic information was analysed, revealing progressive enrichment of B6-specific reads (originating from Xi) upon *Xist* induction for H4K20me1 as is the case for H3K27me3 (Fig EV5C). We analysed relative *B6*-read enrichment within 10-kb windows across the whole X chromosome normalised to $t = 0$ h (Fig 4B). This revealed H4K20me1 accumulation after 8 h of DOX induction, a time point when H3K27me3 also starts to accrue but significantly later than initial H2AK119Ub enrichment (Zylicz *et al*, 2019). The levels of enrichment for H4K20me1 are lower than for H3K27me3 and seem to reach their plateau earlier. To better visualise the timing of accumulation, we normalised both marks to their effective dynamic range, i.e. to average accumulation after 24 h of DOX induction (Fig 4C). This confirmed that after 8 h of DOX treatment, there is concurrent initiation of H4K20me1 and H3K27me3 accumulation. To quantify this further, we plotted the B6-read enrichment relative to $t = 0$ h of each 10-kb window as a function of time and fitted a sigmoidal curve (see Materials and Methods). To extract the information about relative timing of histone mark accumulation, we obtained the time when each curve reaches its maximum slope (effective dose 50%, ED50). ED50 analysis revealed that H4K20me1 reaches its most efficient accumulation prior to H3K27me3 (Fig 4D) but later than H2AK119Ub (Zylicz *et al*, 2019). This is in line with H4K20me1 achieving its plateau significantly before H3K27me3. Thus, our analysis confirmed live imaging observations that H4K20me1 accumulates concurrently with H3K27me3 but quickly reaches maximum enrichment (Fig 3C).

We next examined the degree to which the distributions of the two marks overlapped across the Xi. We found that H4K20me1 becomes preferentially enriched at loci in proximity to the *Xist* gene (green bar) as well as at regions that *Xist* RNA first interacts with (so called "entry sites", black bars (Pinter *et al*, 2012); Fig 4E). The initial enrichment of H4K20me1 on the *Xist*-coated X follows the same pattern previously detected for not only H3K27me3 but also PRC1-dependent H2AK119Ub (Fig 4E; Zylicz *et al*, 2019). To confirm this observation, we investigated chromosome-wide correlation between H3K27me3 and H4K20me1 accumulation within different genomic windows (Fig 4F). Consistently with our initial observation, we found striking correlation in the accumulation of both marks across intergenic windows as well as bodies of silent genes. In contrast, the bodies of genes that were active prior to DOX treatment (i.e. initially active genes) showed a much lower Pearson's correlation with $\rho = 0.2$, implying that both marks differ in their correlation with transcription (see below). With the exception of initially active genes, nChIP-seq of H4K20me1 revealed that its accumulation follows a strikingly similar pattern and dynamics to H3K27me3.

## H4K20me1 accumulates intergenically and is dispensable for XCI initiation

The above nChIP-seq analysis revealed similarities and differences between H4K20me1 and H3K27me3 accumulation on the X during

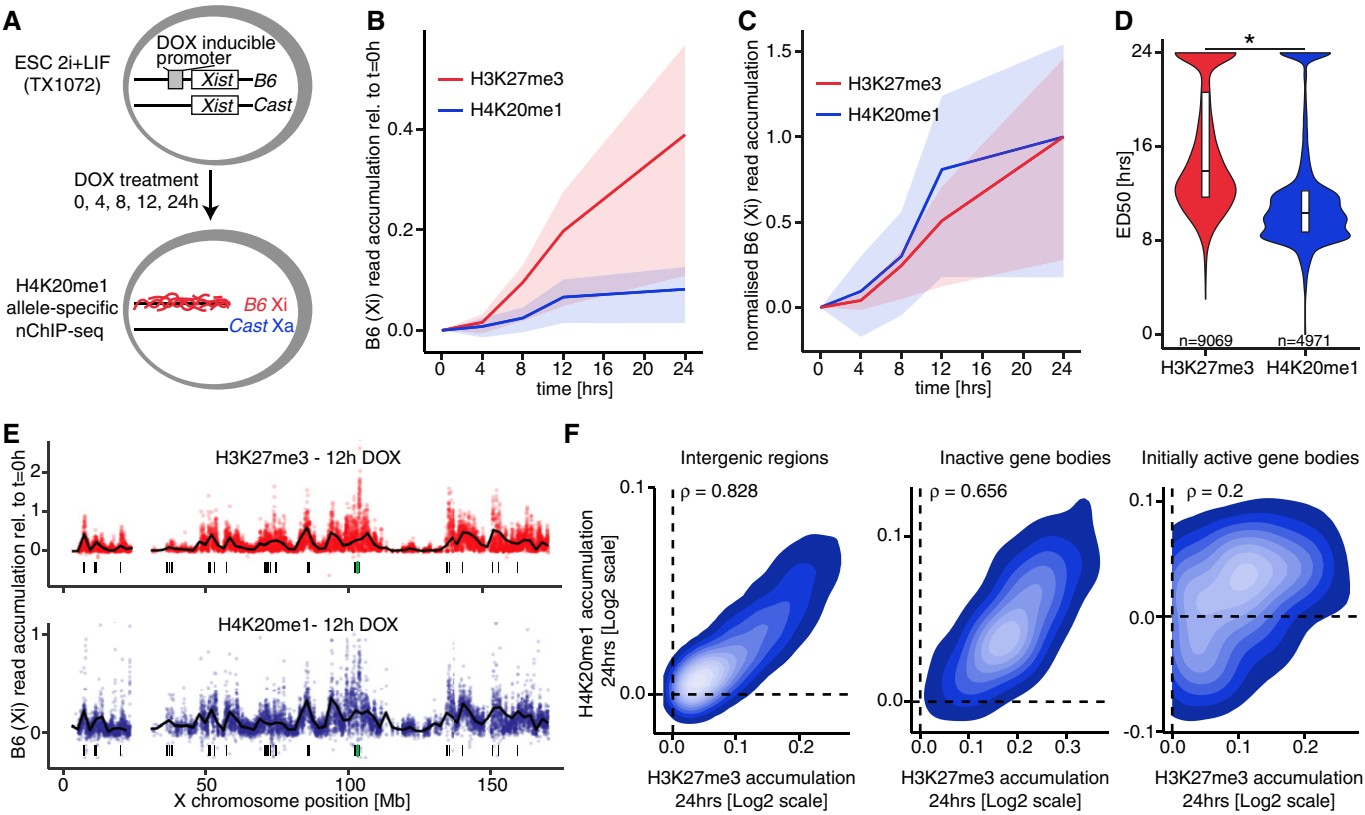

**Figure 4. Native ChIP-seq reveals H4K20me1 and H3K27me3 co-accumulation during XCI.**

A  Schematic representation of the experimental design. The hybrid TX1072 mouse female ESC line was used, in which *Xist* can be induced from the endogenous B6 allele. Time course nChiP-seq for H4K20me1 was performed and compared to a published H3K27me3 dataset (GSE116480) (Zylicz *et al*, 2019).
B  Quantification of average H4K20me1 (blue) and H3K27me3 (red) enrichment at the Xi (B6 allele) compared with *t* = 0 h in 10-kb windows spanning the whole chromosome. Shading is the interquartile range.
C  As in (B) but shown is normalised B6 enrichment to the average accumulation at 24 h.
D  Pairwise comparison of H4K20me1 and H3K27me3 accumulation dynamics (ED50) at the X chromosome. All 10-kb windows with ED50 < 24 h are plotted. * aired Wilcoxon rank-sum test *P*-value < 0.05.
E  H4K20me1 (blue) and H3K27me3 (red) accumulation across the Xi after 12 h of DOX treatment. The black line is a locally estimated scatterplot smoothing (LOESS) regression on all 10-kb windows (dots). Below each plot shown is the *Xist* locus (green bar) and *Xist* entry sites (black bars).
F  Correlation between H3K27me3 and H4K20me1 accumulation after 24 h of DOX treatment at initially active gene bodies (left), inactive gene bodies (middle) and 10-kb intergenic windows spanning the X chromosome (right). All scales are logarithmic. All correlations (ρ) are with *P* < 0.01 from Pearson's correlation test.

XCI. While investigating genes that are initially active and then become silenced following *Xist* induction, we observed that H4K20me1 is strongly biallelically enriched (premarked) at transcribed gene bodies, prior to silencing (Fig 5A). This contrasts strikingly with the distribution of H3K27me3, which never premarks initially active genes (Zylicz *et al*, 2019). We confirmed that this H4K20me1 enrichment at transcribed genes is a general feature both on the X chromosome (Fig 5B) and autosomes (Fig EV5D). This is in line with previous reports indicating that H4K20me1 correlates with transcriptional elongation and is enriched at active gene bodies (Beck *et al*, 2012; Kapoor-Vazirani & Vertino, 2014; Veloso *et al*, 2014). Upon *Xist* induction (+ DOX), we found a significant accumulation of H4K20me1 on the Xi at initially active promoters and intergenic regions (Fig 5A and B). This resembles the pattern observed for H3K27me3, with the intriguing caveat that H4K20me1 does not seem to further accumulate at bodies of initially active genes, rather, H4K20me1 levels

remain constant. Thus, H4K20me1 and H3K27me3 accumulation patterns are strikingly similar within intergenic regions but differ in the bodies of initially active genes (Fig 4F). Next, to evaluate the relationship between H4K20me1 accumulation and the process of gene silencing, we separately analysed genes inactivated early and late upon DOX treatment (Fig 5C). This revealed that while H4K20me1 *de novo* accumulation following *Xist* induction is restricted to intergenic and promoter regions, it occurs more efficiently in the proximity of rapidly silenced genes e.g. of *Rnf12* (Fig 5A). All in all, these findings suggest that prior to XCI, H4K20me1 correlates with active transcription (in gene bodies), but following *Xist* induction H4K20me1 becomes enriched *de novo* at regions surrounding genes that are rapidly silenced.

To further explore the potential relationship of H4K20me1 enrichment and H3K27me3 deposition during XCI, we decided to investigate chromatin states in *Xist* mutant cell lines. Current models suggest that H3K27me3 is deposited thanks to a complex

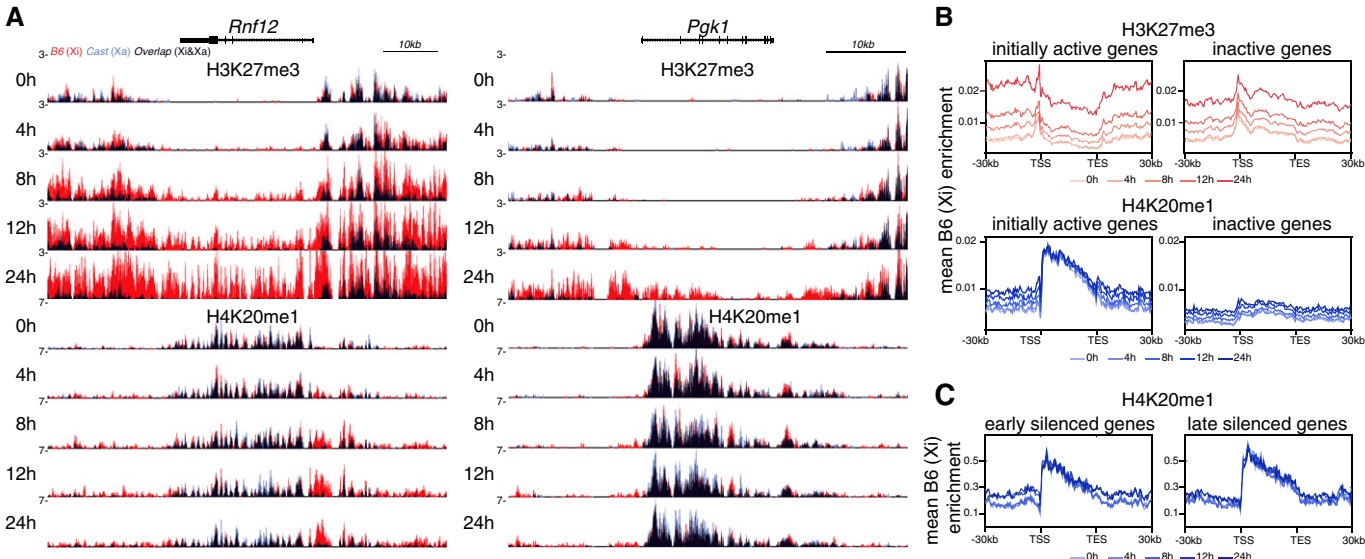

**Figure 5. H4K20me1 accumulates intergenically.**

A   Genome browser tracks showing H3K27me3 (top) and H4K20me1 (bottom) accumulation at a gene silenced rapidly (*Rnf12*) or more slowly (*Pgk1*). Allele-specific tracks were overlaid (B6 in red and Cast in blue). Note strong H4K20me1 bi-allelic premarking at gene bodies.

B   Average H3K27me3 (top) and H4K20me1 (bottom) enrichment at the B6 allele over initially active or inactive genes ± 30 kb at the X chromosome. Shown are data for all time points.

C   Average H4K20me1 enrichment at the B6 allele over early and late silenced genes ± 30 kb at the X chromosome. Shown is data for all time points.

Data information: TSS: transcription start site; TES: transcript end site.

machinery, involving initial PRC1 recruitment by the B and C repeats of *Xist* and subsequent PRC2 recruitment thanks to H2AK119Ub enrichment (Almeida *et al*, 2017; Pintacuda *et al*, 2017; Bousard *et al*, 2019; Colognori *et al*, 2019; Nesterova *et al*, 2019). We therefore tested whether H4K20me1 enrichment at the Xi is also dependent on the B and C regions of *Xist* RNA. To this end, we employed previously established male ESC lines harbouring a DOX-inducible, endogenous *Xist* in either its full length form (*XistFL*) or lacking the B and C repeat regions (*XistΔBC*) (Fig 6A), in which H3K27me3 changes on the Xi had already been mapped (Bousard *et al*, 2019). This system recapitulates hallmarks of XCI, namely chromosome-wide *Xist* coating, X-linked gene silencing and heterochromatin formation (Bousard *et al*, 2019). After DOX treatment, we performed IF/RNA FISH experiments to quantify the efficiency of H4K20me1 enrichment at the *Xist* RNA domains (Fig 6B). While the global level of H4K20me1 was comparable in both cell lines, it showed, as predicted, striking variability between cells due to different cell cycle stages. Nevertheless, we observed complete loss of H4K20me1 enrichment from the *Xist* RNA domains when *XistΔBC* was expressed; a result reminiscent of what was previously observed for H3K27me3 (Fig 6C) (Bousard *et al*, 2019). These data indicate that H4K20me1 enrichment depends on the same regions of *Xist* RNA that are involved in recruiting PRC1 and lead to subsequent PRC2 recruitment. Which factors directly allow for H4K20me1 enrichment at the Xi still remains unclear, however. Another important conclusion from this result pertains to gene silencing. Indeed, the *XistΔBC* RNA is able to initiate gene silencing, albeit at slightly lower efficiency (Bousard *et al*, 2019). This implies that the *de novo* accumulation of H4K20me1 is largely dispensable

for the initiation of gene silencing, similarly to both Polycomb-associated H3K27me3 and H2AK119Ub.

# Discussion

We report the spatio-temporal dynamics of two early chromatin changes, H4K20me1 and H3K27me3 during the formation of facultative heterochromatin in XCI. Using a unique combination of live-cell imaging and chromatin profiling in the same cell systems, we reveal the relative timing and distributions of chromatin enrichment for these marks during X inactivation. Our study provides insights into the process of XCI and new tools for the study of epigenetic processes in general. Indeed, we use the powerful technology of mintbodies, genetically encoded fluorescent probes that can detect specific proteins and their modifications. In this way, we visualise the distribution and levels of specific histone modifications, in longitudinal single-cell analyses of the epigenetic process of XCI. We previously reported on the development of mintbodies specific for H4K20me1 and H3K9ac (Sato *et al*, 2013; Sato *et al*, 2016). The H3K27me3 mintbody developed here will enable the study of PRC2-dependent epigenetic mechanisms beyond XCI.

Our results reveal that first detectable enrichment of both H3K27me3 and H4K20me1 on the *Xist*-coated X chromosome occurs about 45 min following *Xist* RNA coating, thus ~ 3–7 h after DOX treatment. This suggests that both marks become enriched on the Xi in a *Xist*-dependent but likely indirect mechanism. It should be noted that proteins such as SPEN which are recruited directly by *Xist* RNA (Chu *et al*, 2015; McHugh *et al*, 2015; Minajigi *et al*, 2015)

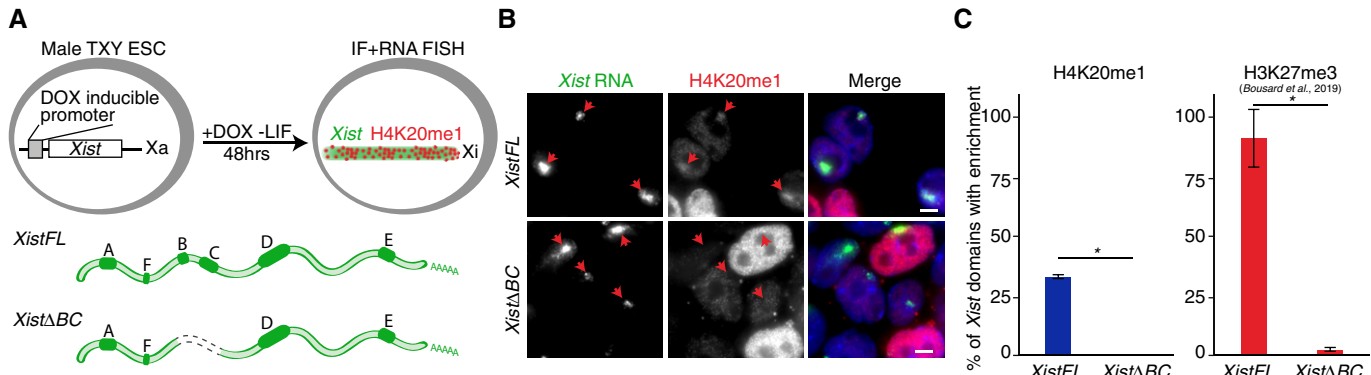

**Figure 6. H4K20me1 accumulation relies on *Xist B + C* repeat region.**

A  Schematic representation of the experimental design. By DOX addition, male TXY mouse ESC lines allow for expression of full length *XistFL* or *XistΔBC*. Cells induced for 48 h in the absence of LIF were used for combined IF and RNA FISH.

B  Representative image of IF/RNA FISH for *Xist* RNA and H4K20me1 in cells expressing *XistFL* or *XistΔBC*. Arrowheads point to the *Xist* RNA domains. Scale bar = 5 μm.

C  Graph represents the mean % ± StDev of *Xist* RNA domains enriched for H4K20me1 (left) or H3K27me3 (right) in cells expressing *XistWT* or *XistΔBC*. Shown are averages from at least 2 independent experiments; minimum of 50 *Xist* RNA domains were counted per experiment; only *P*-values corresponding to significant differences from unpaired Student's *t*-test comparing mutants to *Xist* FL are indicated as * (*P*-value < 0.05). Data for H3K27me3 was extracted from published data (Bousard *et al*, 2019).

show immediate colocalisation with *Xist* during initiation of XCI based on live-cell imaging using the same ESC system (Dossin *et al*, 2020). While the initial enrichment of both H4K20me1 and H3K27me3 follows similar dynamics, they soon diverge with H3K27me3 accumulating more efficiently. What is more, H4K20me1 shows a unique cell cycle dynamics with transiently increased levels during G2/M (Rice *et al*, 2002). In line with these differences, our nChIP-seq analysis revealed that both marks also show distinct genomic distribution especially in the context of initially active genes. Thus, while accumulation of both marks depends on a common *Xist*-mediated mechanism their spreading and long-term enrichment is regulated independently.

Here, we reveal that both H3K27me3 and H4K20me1 accumulation depends on the *Xist* B + C region (Bousard *et al*, 2019). However, neither SETD8 nor PRC2 core components have been identified as direct *Xist* RNA-binding proteins (Chu *et al*, 2015; McHugh *et al*, 2015; Minajigi *et al*, 2015). Instead, PRC1 is thought to be recruited by this region via the hnRNPK factor. Furthermore, SETD8 does not interact with any of the five proteins, including hnRNPK, recently identified to bind with the *Xist* B + C repeats (Bousard *et al*, 2019). It thus remains unclear how SETD8, if at all, is recruited to the Xi and whether it depends on the non-canonical PRC1 complexes, as proposed for PRC2 (Almeida *et al*, 2017; Pintacuda *et al*, 2017). Indeed, PRC2 recruitment depends on the non-canonical PRC1, which very rapidly deposits H2AK119Ub. This modification in turn is proposed to recruit PRC2 via its cofactors, e.g. JARID2 (Cooper *et al*, 2016). Whether a similar mechanism mediates H4K20me1 enrichment remains unclear. A factor potentially involved in indirect SETD8 recruitment is a Polycomb-group protein L3MBTL1, which is not however a cofactor of either PRC1 or PRC2. L3MBTL1 is a binder of H4K20me1 (Kalakonda *et al*, 2008) but it also interacts with SETD8 and H3K27me1/2 (Kalakonda *et al*, 2008). Thus, L3MBTL1 may recruit SETD8, when H3K27 starts to become methylated by PRC2 but the H3K27me1/2 enrichment during XCI has not yet been studied. In line with this model, loss of

EED, a core PRC2 component, has previously been reported to result in partially reduced H4K20me1 enrichment at the Xi (Schoeftner *et al*, 2006). Thus, PRC2 contributes to H4K20me1 accumulation but other pathways must also operate. Alternatively, it is possible that SETD8 is not recruited to the Xi; instead, it could be the enzymes catalysing H4K20me2 demethylation to H4K20me1, which become enriched during XCI. Indeed, in *C. elegans* dosage compensation, which consists of a reduction of X-linked gene activity in XX individuals, is partly dependent on an H4K20me2-specific demethylase (Brejc *et al*, 2017). In this system, efficient conversion of H4K20me2 to H4K20me1 at the X chromosome by DPY-21 promotes gene silencing during the maintenance phase. RSBN1 and RSBN1L are mouse orthologs of DPY-21 and seem to retain specific histone demethylase activity (Brejc *et al*, 2017). Whether RSBN1/RSBN1L have a role in H4K20 methylation dynamics in mammalian cells and whether these factors play a role in XCI merits further investigation. Finally, it is also possible that H4K20me1 enrichment at the Xi is due to hindered conversion to higher methylation states by SUV4-20H1/2.

Identifying the B and C repeats of *Xist* RNA as the key region for H4K20me1 enrichment during XCI suggests that this mark may only play a minor role in initiating gene silencing. Indeed, cells expressing *Xist:ΔBC* can induce X chromosome gene silencing during XCI albeit with lower efficiency (Bousard *et al*, 2019). The fact that H4K20me1 is highly enriched at transcribed gene bodies both on the X chromosome (before XCI) and autosomes would also suggest that this mark is not involved in transcriptional repression. Indeed, previous studies reported a positive correlation between H4K20me1 enrichment and the rate of transcriptional elongation (Veloso *et al*, 2014). SETD8 was also proposed to facilitate RNA Polymerase II (RNAPII) release from promoters (Kapoor-Vazirani & Vertino, 2014; Nikolaou *et al*, 2017). The initial distribution of H4K20me1 is in line with such findings. However, this mark also persists at genes which become rapidly silenced during XCI, e.g. *Rnf12* (Fig 5A). Thus, transcriptional silencing does not result in rapid H4K20me1 depletion at

gene bodies indicating that SETD8 is not directly recruited by the RNAPII. The fact that H4K20me1 remains in gene bodies even as silencing occurs and that it accumulates in intergenic regions during XCI suggests that it may play a rather different role. Indeed, such a mark may be coupled both to transcriptional elongation and to gene silencing in the context of XCI, if it plays a role in genome stability and/or chromatin structure (Schotta *et al*, 2008), rather than in gene transcriptional regulation *per se.*

While H4K20me1 is dispensable for the initiation of gene silencing, it might participate in chromatin compaction during XCI. Indeed, DNA FISH analysis of cells depleted of SETD8 previously revealed some decompaction of the Xi (Oda *et al*, 2009). The finding that H4K20me1 predominantly accumulates intergenically upon *Xist* induction might support this conclusion. Consistently, in *C. elegans* loss of H4K20me1 enrichment at the X chromosome leads to a decrease in long-range interactions, suggesting impaired chromatin compaction (Brejc *et al*, 2017). How H4K20me1 might mediate such a process is still far from clear. One factor likely to be involved is L3MBTL1, which becomes recruited to H4K20me1 during S-phase (Trojer *et al*, 2007; Kalakonda *et al*, 2008). *In vitro*, L3MBTL1 compacts nucleosomes in a H4K20me1/2-dependent manner (Trojer *et al*, 2007). However, the involvement of L3MBTL1 in XCI and chromatin compaction remains to be defined.

In conclusion, we have applied a unique combination of mintbodies to track PRC2-dependent H3K27me3 and H4K20me1 in living cells, together with a high resolution ChIP-seq analysis to define the precise spatio-temporal distribution of these chromatin marks across the X chromosome during the initiation of XCI. Our approach has uncovered that H3K27me3 and H4K20me1 become enriched thanks to the same B + C region of *Xist* RNA but that the subsequent accumulation dynamics of these marks is rather different during XCI. While *de novo* H4K20me1 enrichment is dispensable for the initiation of gene silencing, it might contribute to chromatin compaction during XCI. These observations may have implications for the more general chromatin function of H4K20me1 across the genome.

# Materials and Methods

### Cloning variable fragments of H3K27me3-specific antibody and mutagenesis

Mouse hybridoma clone 2E12 was generated and screened by Mab Institute Inc. as described (Kimura *et al*, 2008). RNA was extracted from cells using TRIzol (Thermo Fisher Scientific) for RNA-seq and de novo transcriptome assembly to determine cDNA sequence encoding the IgG heavy and light chains, according to Kuniyoshi *et al* (2016). To amplify variable regions of heavy ($V_H$) and light ($V_L$) chains, PrimeScript II High Fidelity One step RT-PCR Kit (Takara) was used with the following primers: VH_s, CGAATTCGC CATGGCCCAGGTCCAGTTGCAGCAGTCT; VH_as, TGAACCGCCTC CACCTGAGGAGACTGTGAGAGTGGT; VL_s, TCTGGCGGTGGCGGA TCGGACATTGTGATGTCACAGTCT; VL_as, TGGATCCGCCCGTTT GATTTCCAGCTTGGTGCCTCC. As described previously (Sato *et al*, 2013), the heavy and light chains were connected by PCR (seven cycles) using the amplified fragments with the following oligonucleotides to have a flexible linker: LINK primer1, GTCTCCTCAGGTG GAGGCGGTTCAGGCGGAGGTGGCTCTGGCGGTGGCGGATCG; and

LINK primer2, CGATCCGCCACCGCCAGAGCCACCTCCGCCTGAACC GCCTCCACCTGAGGAGAC. The resulting scFv fragments were further amplified (20 cycles) using the 5′ and 3′ primers: scFv primer_s, CTCGAGCTCAAGCTTCGAATTCGCCATGGCCCAGG and scFv primer_as: CATGGTGGCGACCGGTGGATCCGCCCGTTTTATTT CCAG, for cloning into psfGFP-N1 vector (Addgene #54737) digested with EcoRI and BamHI to yield psfGFP-N1-2E12 using In-Fusion (Takara).

Error-prone PCR was performed to make a few amino acid substitutions in the scFv region using psfGFP-N1-2E12 as a template and a set of primers (scFv primer_s and scFv primer_as) with Taq DNA Polymerase (QIAGEN) in the presence of $MnCl_2$ (six cycles) (Wilson & Keefe, 2000).

Site-directed mutagenesis was performed by inverse PCR using a mutant clone that contained M86L mutation (psfGFP-N1-2E12M86L) as a template with PrimeSTAR HS DNA polymerase (Takara) and primers to introduce M185I substitution: a472t_s, TGTCAGTTGGA GAGAAGGTTACTTTGGCCTGCAAGT; and a472t_as, ACTTGCAGGC CAAAGTAACCTTCTCTCCAACTGACA.

### Mintbody expression in HeLa and MC12 cells

HeLa and MC12 cells were described previously (Sato *et al*, 2016). Transfection was performed using Fugene HD (Promega) or Lipofectamine 2000 (Thermo Fisher Scientific) according to the manufacturer's instructions. To obtain stably expressing cells, scFv-sfGFP was cloned into PB533A (System Biosciences) and transfected with the transposase expression vector (PB200PA-1; Systems Biosciences). The day after transfection, cells were grown in 1 mg/ml G418 (Nacalai Tesque) for 1–2 weeks, and fluorescent cells were collected using a cell sorter (SH800; Sony). Single-cell clones were obtained by limiting dilution into 96-well plates and microscopic investigation.

For live-cell imaging, cells were plated onto a glass bottom dish (Mat-Tek; P35G-1.5-14-C), or a glass bottom plate (AGC Techno Glass; 24-well or 96-well) in FluoroBrite DMEM (Thermo Fisher Scientific) containing 10% FCS (Thermo Fisher Scientific) and 1% Glutamine-Penicillin-Streptomycin solution (Sigma-Aldrich). A glass bottom dish or a plate was set on to a heated stage (Tokai Hit) with a $CO_2$ control system (Tokken) on an inverted microscope (Nikon Ti-E) or on a confocal system (Nikon Ti-E with Yokogawa CSU-W1 spinning disk or Olympus FV1000) to maintain cells at 37°C under 5% $CO_2$ atmosphere.

For immunofluorescence, MC12 cells expressing H3K27me3-mintbody (2E12LI-sfGFP) were fixed with 4% formaldehyde (Electron Microscopy Sciences) in 250 mM HEPES-NaOH (pH 7.4; Wako Purechemicals) containing 0.1% Triton X-100 for 10 min at room temperature. Fixed cells were permeabilised with 1% Triton X-100 in PBS for 20 min at room temperature and washed three times with PBS. After blocking with 10% Blocking One-P (Nacalai Tesque) in PBS for 20 min at room temperature, cells were incubated in 2 µg/ml Cy3-conjugated anti-H3K9me3 (CMA318/2F3; Hayashi-Takanaka *et al*, 2011) and 2 µg/ml Cy5-conjugated anti-H3K27me3 (CMA323/ 1E7; Hayashi-Takanaka *et al*, 2011) for 2 h at room temperature. After washing three times with PBS, cells were incubated with 1 µg/ ml Hoechst33342 in PBS for 20 min at room temperature. Fluorescence images were sequentially collected using an Olympus FV1000 equipped with a 60× PlanApoN (NA 1.40) oil-immersion lens.

**Bacterial expression and purification of mintbody**

2E12LI-sfGFP was subcloned into pMAL-c5X (New England Biolabs) to obtain an MBP-2E12LI-sfGFP expression vector (pMAL-c5X-2E12LI-mintbody). *Escherichia coli* ER2523 cells (NEB Express; New England Biolabs) harbouring the pMAL-c5X-2E12LI-mintbody plasmid were grown in 2 ml Plus grow medium (Nacalai Tesque) containing 100 µg/ml ampicillin at 26°C overnight. The whole 2 ml overnight culture was added to 50 ml Plus grow medium, and cells were grown at 26°C for 1 h before the addition of 0.1 mM isopropyl-β-d-thiogalactopyranoside (IPTG). Cells were further grown for 8 h at 26°C and collected by centrifugation (4,000 × *g*; 10 min; 4°C), and the pellet was stored at −30°C. After thawing on ice, the pellet was resuspended in 2.5 ml of Column Buffer (20 mM Tris–HCl, pH 8.0, 200 mM NaCl, 1 mM EDTA), and cells were disrupted using a sonicator (Branson; Sonifier 250; Output 1.2, Duty Cycle 30%, 4 min with intervals on ice). After centrifugation (2,000 *g*; 10 min; 4°C), the supernatant was collected, diluted 7-fold with Column Buffer and mixed with 200 µl amylose resin (New England Biolabs) that were pre-equilibrated with Column Buffer. After washing with 15 ml Column Buffer, bound proteins were eluted with 1.5 ml Column Buffer containing 20 mM maltose. To remove MBP moiety, the concentration of eluted MBP-mintbody was adjusted to 1 mg/ml and digested with 30 µg/ml Factor Xa (New England Biolabs) for 24 h. The cleavage was confirmed by 10–20% SDS-polyacrylamide gel electrophoresis (SDS–PAGE). After the buffer was replaced with 20 mM sodium phosphate (pH 5.5) buffer containing 25 mM NaCl using a PD MiniTrap G-25 column (GE Healthcare), the digested sample was applied to a HiTrap Q HP column and every 1.5 ml was collected during the application and washing with the same buffer. Fractions containing H3K27me3-mintbody (2E12LI-sfGFP), analysed by 10–20% SDS–PAGE, were concentrated using an Amicon Ultra filter unit (Merck; 10 k cut-off).

For peptide array analysis, a MODified Histone Array (Active Motif) was blocked for 1 h at room temperature in Blocking One (Nacalai Tesque) and incubated for 3 h with 0.4 µg/ml H3K27me3-mintbody in Can Get Signal (Toyobo). After washing three times for 5 min in Tris-buffered saline (20 mM Tris–HCl, pH 8.0, 150 mM NaCl) containing 0.1% Triton X-100 (TBST), the peptide array was incubated for 1 h with horseradish peroxidase-conjugated anti-GFP (Anti-GFP pAb-HRP-DirecT; 1: 2,000 dilution; MBL) and washed 3 times for 10 min in TBST. Signals were developed using Western Lightning Plus-ECL (PerkinElmer) and detected using a LuminoGraph II (ATTO).

For immunofluorescence, HeLa cells were plated in a 24-well glass bottom plate (IWAKI) and transfected with expression vectors for HaloTag-tagged KDM6B and KDM4D. Twenty-four hours after transfection, the cells were fixed, permeabilised and blocked as described above. The cells were incubated for 1 h at room temperature with 5 µg/ml H3K27me3-mintbody, 2 µg/ml Alexa Fluor 488-conjugated anti-H3K27me3 (CMA323/1E7), or 2 µg/ml Alexa Fluor 488-conjugated anti-H3K9me3 (CMA318/2F3) in 10% Blocking One-P (Nacalai Tesque) containing 1 µg/ml Hoechst33342 and Janelia Fluor 646 HaloTag Ligand (a gift from Luke Lavis). After washing three times with PBS, fluorescence images were sequentially collected using Nikon Ti-E equipped with a 40 × PlanApo (NA 0.9) lens. The nuclear intensities of fluorescence signals were quantified using NIS-elements analysis software.

**sgRNA-dCas system to label X-linked loci**

To search the target microsatellite repeats for dCas9-GFP-mediated genome visualisation, we first enumerated the sequences of $N_{20}GG$ and $N_{21}GG$ whose specificities in a chromosome are greater than 19. The specificity of a sequence $s$ in chromosome $c$ is defined to be the ratio $f_c(s)/1 + \sum_{i \neq c} f_i(s)$, where $f_c(s)$ denotes the frequency of $s$ in $c$. Since this entails that sequences appearing less than 20 times in all chromosomes are irrelevant, we employed the $k$-mer counting software DSK (version 2.1.0; (Rizk *et al*, 2013)) to obtain a list of 22- and 23-mers with a copy number > 19 on a chromosome. These sequences and their complementary sequences were used as inputs to the alignment software Bowtie (version 1.1.2; (Langmead *et al*, 2009)) against the FASTA files (GRCm38, Ensembl 84). We next searched for candidate sequences enriched > 19 times in less than 1 Mb. Let $R$ be a region in $c$, then the frequency of $s$ in $R$ is $f_c(s) − n_R(s)$ for some integer $n_R(s)$. Therefore, the condition that the specificity of $s$ in $R$ being > 19 results in $n_R(s) \leq \left( f_c(s) - 20\sum_{i \neq c} f_i(s) - 20 \right)/21$. For sequence $s$ that satisfied the criteria, a comprehensive search for connected regions of lengths less than 1 Mb containing $f_c(s) − n_R(s)$ alignments of $s$ were performed on its Bowtie alignment output. Among the candidates, 32 sequences were experimentally screened.

The expression vectors for sgRNA and dCas9-EGFP were constructed using those obtained from Yolanda Markaki and Heinrich Leonhardt (Anton *et al*, 2014). pex-A-U6p-sgRNA(F + E)-BbsI was first generated from pEX-A-U6-sgRNA-antiMiS, which contained microsatellite-specific sequence, to clone X-linked sgRNA into the BbsI site. Oligonucleotides consisting of the target sequence and BbsI sites were synthesised (Hokkaido System Science; Table EV1) and annealed to insert into the linearised pex-A-U6p-sgRNA(F + E)-BbsI. Each of the resulting plasmids was cotransfected with dCas9-EGFP expression vector into MC12 cells grown on a 24-well glass bottom plate (IWAKI) using Lipofectamine 2000 (Thermo Fisher Scientific), according to the manufacturer's instruction. Among 32 sgRNA screened, 7 that exhibited distinct spots by transient expression (Table EV1) were subcloned into a PiggyBac vector PB510B-1 (System BioSciences) to establish stable lines. dCas9-EGFP was also subcloned into another PiggyBac vector PB533-A2 (System BioSciences). The resulting PB510B-based sgRNA expression vector and PB533-dCas9-EGFP were cotransfected with a transposase expression vector PB200PA-1 (System BioSciences) into MC12 cells, and stable clones were selected in 1 mg/ml G418 and 1 µg/ml puromycin.

The selected MC12 cell lines were plated on a 35-mm glass bottom dish (IWAKI) and fixed with 4% formaldehyde (Electron Microscopy Sciences) in 250 mM HEPES-NaOH (pH 7.4; Wako Purechemicals) containing 0.1% Triton X-100 for 5 min at room temperature. Fixed cells were permeabilised with 1% Triton X-100 in PBS for 20 min at room temperature and washed three times with PBS. The cells were incubated with 2 µg/ml Cy5-conjugated anti-H3K27me3 (CMA323/1E7; Hayashi-Takanaka *et al*, 2011) and 1 µg/ml Hoechst33342 in 10% Blocking One-P (Nacalai Tesque) in PBS for 1 h at room temperature. After washing three times with PBS, fluorescence images were sequentially collected using a confocal microscope (Nikon Ti-E with Yokogawa CSU-W1 spinning disk and Andor iXon3 EM-CCD).

To obtain mitotic cells, MC12 cells were plated on 35-mm glass bottom dish (IWAKI) and treated with colcemid (Nacalai Tesque) at a final concentration of 50 ng/ml in the medium for 90 min. After

removal of the medium, 75 mM KCl was added to the dish and incubated for 30–40 min at room temperature to swell the cells. The mitotic cells were fixed with 250 mM HEPES-NaOH (pH 7.4; Wako Purechemicals) for 5 min at room temperature and permeabilised with 0.1% Triton X-100 in PBS for 20 min at room temperature. After washing with PBS, cells were stained with Cy5-conjugated anti-H3K27me3 and Hoechst33342, as described above.

To visualise X-linked loci simultaneously with H3K27me3 or H4K20me1 during X inactivation, female mESC lines expressing the two sgRNA (mX8 and mX26), 3×sfGFP-dCas9 and H3K27me3-mintbody (SNAP-Tag version, visualised with JF646 (Grimm et al, 2015) and H4K20me1-mintbody (mCherry version) were established. For tracking differentiating mouse ESCs in 3D over several hours, we used mintbodies tagged with nuclear localisation signals to increase the brightness in nuclei. Therefore, the oscillation of the global modification levels throughout the cell cycle was not monitored (Rice et al, 2002). PGK12.1 mouse female ES cells (Penny et al, 1996) were routinely grown on dish coated with 0.1% gelatin (Wako Purechemicals) in DMEM high glucose (Nacalai Tesque) with 10% FBS (Thermo Fisher Scientific), 1% L-Glutamine–Penicillin–Streptomycin Solution (Sigma-Aldrich), 1% MEM non-essential amino acid solution (Nacalai Tesque), 1 mM Sodium Pyruvate (Thermo Fisher Scientific), 55 μM 2-mercaptoethanol (Thermo Fisher Scientific), $10^3$ units/ml leukaemia inhibitory factor (LIF; Nacalai Tesque) at 37°C and 5% $CO_2$ atmosphere. PGK12.1 cells were transfected with the expression vectors using Lipofectamine 2000 (Thermo Fisher Scientific) and selected in 1 mg/ml G418 and 1 μg/ml puromycin to obtain stable cell lines. To induce differentiation, cells were plated at $1–2 \times 10^4$ cells per 10 cm dish and incubated with LIF-free medium. For live imaging, mESCs were plated on 35-mm μ-Dish (ibidi) coated with laminin (BioLamina) in FluoroBrite DMEM (Thermo Fisher Scientific) containing all the other supplements described above. The fluorescence images were sequentially collected using a confocal system (Nikon Ti-E with Yokogawa CSU-W1 spinning disk and Andor iXon3 EM-CCD) with a heated stage (Tokai Hit) to maintain cells at 37°C under 5% $CO_2$ atmosphere.

## TX-based ESC lines

TX1072 [mouse, female, (Mus musculus castaneus X C57BL/6) embryonic stem cells] cells have been previously derived in the lab (Schulz et al, 2014). TX-BglSL cell line has been previously derived from TX1072 (Dossin et al, 2020). TX-Xist-EGFP was established by targeting a BglG-EGFP expression cassette into the ROSA26 permissive locus. This was done by simultaneously transfecting TX-BglSL ESCs with a targeting vector (ROSA26 homology arms, CAGS promoter and BglG-EGFP-NLS fusion gene) and two pX330 plasmids containing two sgRNAs to ROSA26 (TGGGCGGGAGTCTTCTGGGC and ACTGGAGTTGCAGATCACGA). TX-Xist-mCherry cell line was derived by targeting BglG-mCherry expression cassette into the TIGRE permissive locus. This was done by simultaneously transfecting TX-BglSL ESCs with a targeting vector (TIGRE homology arms, CAGS promoter and BglG-mCherry-NLS fusion gene) and a single pX459 plasmids containing an sgRNA to TIGRE (ACTGCCATAA-CACCTAACTT). After brief puromycin selection, clones were picked and screened by imaging. Subsequently, clones with heterozygous insertion of the BglG cassette were selected by genotyping PCR. The

TX-Xist-EGFP; H4K20me1-mCherry cell line was derived by cotransfecting TX-Xist-EGFP with pBaso transposase vector and a PiggyBac vector harbouring an expression cassette for H4K20me1 mintbody (15F11-mCherry) (Sato et al, 2016). The TX-Xist-mCherry; H3K27me3-sfGFP cell line was derived by cotransfecting TX-Xist-mCherry with pBaso transposase vector and a PiggyBac vector harbouring an expression cassette for H3K27me3 mintbody (2E12LI-sfGFP). All TX1072-based ESC lines were cultured on gelatine-coated plates in Dulbecco's Modified Eagle Medium (DMEM) supplemented with 15% foetal bovine serum, 2-mercaptoethanol (0.1 mM), LIF (1,000 U/ml) and 2i (PD0325901 [0.4 mM], CHIR99021 [3 mM]). Cells were incubated at 37°C with 8% $CO_2$.

TXY:XistFL (mouse, male embryonic stem cells) were obtained from the Wutz team (Wutz et al, 2002), and TXY:XistΔBC was previously derived in the laboratory (Bousard et al, 2019). These lines were cultured on gelatine-coated plates in Dulbecco's Modified Eagle Medium (DMEM) supplemented with 15% foetal bovine serum, 2-mercaptoethanol (50 mM) and LIF (1,000 U/ml). Cells were incubated at 37°C with 8% $CO_2$.

## DOX treatment

For TX1072-based ESC lines after 24 h of culture in DMEM/15% FCS + LIF + 2i, the medium was supplemented with doxycycline (1 μg/ml). For TXY-based ESC lines after 24 h of culture in DMEM/15%FCS + LIF, cells were washed twice with PBS and medium was changed to DMEM/10%FCS-LIF with doxycycline (1.5 μg/ml). Cells were collected after 48 h of differentiation.

## Antibody validation

H3K27me3 mintbody and H4K20me1 antibody used for nChIP-seq (Active Motif cat. 39727) were validated for sensitivity and specificity using the MODified Histone Peptide Array (ActiveMotif) following manufacturer's instructions.

## Native ChIP-seq

Native ChIP-seq was performed as previously described (Zylicz et al, 2019). Cells were collected using Accutase (Thermo Fisher Scientific), washed twice in ice-cold PBS and counted. Typically, 10 million (mln) cells were used per immunoprecipitation (IP). A fraction of cells was always used for RNA/FISH verification of Xist induction. Cell pellet was resuspended in 90 μl (per 10 mln cells) of Lysis Buffer (50 mM Tris–HCl, pH 7.5; 150 mM NaCl; 0.1% sodium deoxycholate; 1% Triton X-100; 5 mM $CaCl_2$; Protease Inhibitor Cocktail (Roche); 5 mM sodium butyrate). After lysing cells on ice for 10 min, we added 62 μl (per 10 mln cells) of Lysis Buffer with MNase (500 μl buffer + 0.5 μl MNase). Chromatin was digested for exactly 8 min at 37°C, and reaction was stopped by the addition of 20 mM EGTA. To remove undigested debris, the lysates were centrifuged at 14,000 g for 5 min at 4°C. Supernatant was transferred to a fresh tube, an equal volume of STOP Buffer (50 mM Tris–HCl, pH 7.5; 150 mM NaCl; 0.1% sodium deoxycholate; 1% Triton X-100; 30 mM EGTA; 30 mM EDTA; Protease Inhibitor Cocktail; 5 mM sodium butyrate) was added, and samples were stored on ice. For the nChIP input, 5 μl of lysate was digested in 45 μl of ProtK Digestion Buffer (20 mM HEPES; 1 mM EDTA; 0.5% SDS, 0.8 mg/

ml Proteinase K) for 30 min at 56°C. 50 µl of AMPure XP beads were added to the digested lysate together with 60 µl of 20% PEG8000 1.25 M NaCl. After mixing, the samples were incubated for 15 min at RT. Beads were separated on a magnet and washed twice with 80% ethanol for 30 s. DNA was eluted in 12 µl of Low-EDTA TE. DNA isolated in this step was used as the input sample. The volume of each undigested lysate was adjusted to obtain 1 ml per IP using a 1:1 mix of Lysis Buffer and STOP Buffer. Anti-mouse Dynabeads (25 µl/IP) were washed twice in Blocking Buffer (0.5% BSA; 0.5% Tween in PBS). Beads were then resuspended in Blocking buffer and coated with antibodies for 4 h at 4°C [H4K20me1 (0.5 µg/IP), Active Motif cat. 39727]. Once coated beads were magnet-separated and resuspended in 1 ml of lysate, samples were left rotating overnight at 4°C. Following day, beads were magnet-separated and washed quickly with ice-cold washing buffers: four times with Low Salt Buffer, two times with High Salt Buffer (0.1% SDS; 1% Triton X-100; 2 mM EDTA; 20 mM Tris–HCl, pH 8.1; 360 mM NaCl; 0.1% sodium deoxycholate) and two times with LiCl buffer (0.25 M LiCl; 1% NP40; 1.1% sodium deoxycholate; 1 mM EDTA; 10 mM Tris–-HCl pH 8.1). Prior to elution, all samples were rinsed once in TE. ChIP-DNA was eluted in ProtK Digestion buffer for 15 min at 56°C. Beads were separated, and the supernatant was further digested for another 2 h at 56°C. DNA was isolated using AMPure XP beads as described for the input sample.

For each nChIP-seq, 0.5 µl of each sample was used for qPCR validation of enrichment at control regions. 0.5 µl of input samples were also used to verify the digestion efficiency using D1000 tapestation. Remaining DNA concentration was adjusted and used for library preparation using Ovation® Ultralow Library System V2 following suppliers protocol. Amplified libraries were size-selected for dinucleotide fraction (350–600 bp fragments) using agarose gel-separation and MinElute Gel Extraction Kit (QIAGEN). Sample quality was inspected using D1000 tapestation. Samples were sequenced using a HiSeq2500 with Paired-End (PE) 100 or a HiSeq4000 PE150.

## nChIP-seq data processing

Adapters and low quality bases (< Q20) have been removed with TrimGalore (v0.4.4; http://www.bioinformatics.babraham.ac.uk/projects/trim_galore) and Cutadapt (1.12) (Martin, 2011). Moreover, for PE150 samples, last 50 pb were trimmed to allow unbiased comparison of the data with PE100 samples. An "N-masked" genome has been generated with SNPSplit (0.3.2) (Krueger & Andrews, 2016) which is a version of the mouse reference genome mm10 where all the polymorphic sites for the hybrid strain *Mus musculus* CAST/EiJ and *Mus musculus* C57BL/6 are masked by ambiguity nucleobase "N". For all samples, reads were then mapped to the "N-masked" genome with STAR (2.5.3a) with options [--outFilterMultimapNmax 1 --outSAMmultNmax 1 --outFilterMismatchNmax 999 --outFilterMismatchNoverLmax 0.03 --alignIntronMax 1 --alignEndsType EndToEnd --outSAMattributes NH HI NM MD] (Dobin *et al*, 2013). Duplicates were discarded with Picard MarkDuplicates (1.65) with options [REMOVE_DUPLICATES=true] (https://broadinstitute.github.io/picard/) and reads mapped on blacklisted regions from Encode Consortium were discarded. SNPSplit (0.3.2) (Krueger & Andrews, 2016) was then used to generate allele-specific BAM files by separating the alignment into two distinct alleles (CAST and B6) based on SNPs information

downloaded from Sanger. Bigwig files were created with bedtools genome CoverageBed (2.27.1) (Quinlan & Hall, 2010), using a scale factor calculated on the total library (10,000,000/total reads) for both allele-specific bigwigs, and loaded on UCSC genome browser. H3K27me3 nChIP-seq data were downloaded from GSE116990 and processed identically to H4K20me1 nChIP-seq.

### Windows definition

Global analysis was first done on fixed windows (10 kb) spanning the whole genome and then on different genomic subcategories: initially active genes, silent genes and intergenic regions. Initially active genes were defined as genes with a transcript having its TSS (refFlat annotation) overlapping a consensus peak of H3K9ac and a consensus peak of H3K4me3. Moreover, initially active genes were filtered to remove potentially bivalent genes by discarding genes with a normalised H3K27me3 signal superior to 0.035 at $t0$, value defined from the intersection of density curves of H3K27me3 premarking signal of genes initially defined as active and inactive. For genes having several active transcripts detected, the active gene was defined as starting at the minimum start of transcripts and ending at the maximum end of transcripts. Moreover, using RNA-seq data (Zylicz *et al*, 2019), genes having a mean TPM (2 samples) < 1 at $t0$ were excluded from this list. Silent genes were defined as genes with TSS not overlapping a consensus peak of H3K9ac and a consensus peak of H3K4me3. Using RNA-seq data (ref), genes having a mean TPM > 1 at $t0$ were excluded from this list. Intergenic regions were defined as 10 kb windows not overlapping a gene (active or inactive) and its promoter (2 kb upstream) or an active enhancer (defined in Zylicz *et al*, 2019).

### Counts and normalisation

For all defined windows, total and allelic reads overlapping those features were then counted using featureCounts (1.5.1), with options [-C -p -P] (Liao *et al*, 2014). Then, analysis was done based on normalised reads from B6 allele (allele of the inactive X chromosome). For each sample, a normalisation factor was calculated with the trimmed mean of $M$-values method (TMM) from edgeR package (Robinson *et al*, 2010), based on B6 reads overlapping consensus peaks located on autosomes. Peaks of each sample were first identified using MACS2 (2.0.10) with -B --broad options. For each replicate, all peaks coordinates were merged using bedtools merge (2.27.1) (Quinlan & Hall, 2010). Then, common regions between merged peaks coordinates of each replicate were selected using bedtools intersectBed (2.27.1) (Quinlan & Hall, 2010) to represent B6 read accumulation compared to time 0, subtraction of normalised initial counts (time 0) was then applied to all other time points.

### Analysis of the dynamics

Sigmoidal fitting of B6 read accumulation in function of time has also been done with the four-parameter log-logistic function from drc R package. Sigmoidal fittings with low residuals (< mean(residuals) + 1.5 sd(residuals)) were selected. The $ED_{50}$ of the sigmoidal fitting were calculated for each window. Windows with $ED_{50}$ derivative superior to 24 h, or not calculated, were considered as windows with a late accumulation of repressive marks, those were then replaced by 24 h. For pairwise comparison between marks, only windows with $ED_{50}$ inferior to 24 h for both marks were selected.

### Average plots over features

Average plots were created around initially active and silent genes using DeepTools (3.0.2) (Ramirez *et al*, 2014). Matrix counts were created using DeepTools computeMatrix around genes (see above) on chrX and autosomes separately (with option [--binSize 500]), and plots were then created using DeepTools PlotProfile.

Average plots were created with same method around early and late silenced genes. Those genes were defined with a three clusters k-means based on IC35 values calculated after sigmoidal fitting of TT-seq data such as described in (Zylicz *et al*, 2019), defining early, intermediate and late silenced genes.

### IF/RNA FISH

IF/RNA FISH experiments were performed as previously (Bousard *et al*, 2019). *Xist* FL and mutant ES cells were differentiated for 48 h in the presence of DOX (1.5 μg/ml) on gelatine-coated 22 × 22 mm coverslips. Cells were fixed in 3% PFA in PBS for 10 min at RT, followed by permeabilisation in PBS containing 0.5% Triton X-100 and ribonucleoside-vanadyl complex (New England Biolabs) on ice for 5 min. After three rapid washes in PBS, samples were blocked for, at least, 15 min with 5% gelatine from cold water fish skin (Sigma) in PBS. Coverslips were incubated with the H4K20me1 primary antibody (Abcam, abb9051) diluted in blocking solution in the presence of a ribonuclease inhibitor (0.8 μl/ml; Euromedex) for 45 min at RT. After three washes with PBS for 5 min, the coverslips were incubated with a secondary antibody (anti-rabbit antibodies conjugated with Alexa fluorophores diluted 1:500) for 45 min in blocking solution supplemented with ribonuclease inhibitor (0.8 μl/ml; Euromedex). Coverslips were then washed three times with PBS for 5 min at RT. Afterwards, cells were post-fixed with 3% PFA in PBS for 10 min at RT and rinsed three times in PBS and twice in 2 × SSC. Excess of 2 × SSC was removed, and cells were hybridised with a *Xist* p510 probe labelled with Alexa labelled dUTPs. After the RNA FISH procedure, nuclei were stained with DAPI (Sigma-Aldrich), diluted 1:5,000 in 2 × SCC for 5 min at RT and mounted with Vectashield Mounting Medium (Vectorlabs). Cells were imaged with a widefield fluorescence microscope Zeiss Axio Observer (Carl Zeiss MicroImaging) with a 63× oil objective using the filter sets FS43HE, FS38HE, FS50 and FS49. Digital images were analysed with the FIJI platform (https://fiji.sc/). Enrichment of the H4K20me1 signals over *Xist* cloud marked by RNA FISH was counted from at least 50 cells per single experiment.

### RNA extraction, reverse transcription, pyrosequencing

RNA was extracted according to the manufacturer's recommendations using RNeasy Mini Kit (QIAGEN) with on-column DNase digestion (QIAGEN). For cDNA synthesis 1.1 μg RNA was reverse-transcribed using Superscript III Reverse Transcriptase (Thermo Fisher Scientific). For allelic-skewing analysis, the cDNA was PCR-amplified with biotinylated primers and sequenced using the Pyromark Q24 system (QIAGEN).

### Live-cell imaging of TX1072-based ESC lines

For all TX1072-based ESC lines one day before imaging, 30,000 cells were seeded in a fibronectin coated μ-Slide 8 well with glass bottom (Ibidi) in 300 μl medium. To induce *Xist* expression, doxycycline was added to medium 5 min before imaging. Live-cell imaging was performed on an inverted spinning disk confocal microscope Roper/Nikon using a Evolve EM-CCD camera (Photometrics) with a 60× oil objective. Images were taken every 15 min for 10 h with Z slices with 400-nm intervals. During imaging, cells were kept in a chamber at 37°C and 5% $CO_2$.

Quantification of mintbody accumulation was performed using Icy Version 1.9.8.1 and ImageJ. Quantification was done on maximum projections of z-planes. The TX-*Xist*-EGFP or TX-*Xist*-mCherry channel was segmented into an *Xist*-enriched region to separate the Xi from the rest of the nucleus. This region was then applied to the mintbody channel, and mean intensity of the signal was measured per time point. To normalise for the background signal in the nucleus, every 10 time points the signal in the same nucleus was measured using ImageJ. This was used to calculate the background for every time point. The value of signal in the cloud was then divided by this background.

## Data availability

The accession number for the sequencing datasets reported in this paper is GEO: GSE153146 (http://www.ncbi.nlm.nih.gov/geo/query/acc.cgi?acc=GSE153146). Nucleotide sequence data of 2E12LI scFv are available in several public databases (DDBJ/EMBL/GenBank) under the accession number LC597262.

**Expanded View** for this article is available online.

## Acknowledgements

We are grateful to members of the Heard Lab (Curie Institute and EMBL Heidelberg), especially Francois Dossin, and members of the Kimura Lab (Tokyo Tech) for their help and critical input to this project. We also want to thank the Bioimaging facility at iMM JLA for their technical support with fluorescent light microscopy and imaging analysis, and the Biomaterials Analysis Division, Open Facility Center, Tokyo Institute of Technology for DNA sequencing analysis. We thank Yolanda Markaki and Heinrich Leonhardt (LMU Munich) for sgRNA-dCas9 plasmid constructs and Luke Lavis for JF646 dye. We thank Claus S. Sørensen for sharing H4K20me2 antibodies. This work was supported by Fundação para a Ciência e Tecnologia (S.T.d.R), project grants PTDC/BIA- MOL/29320/2017 IC&DT (A. C. R. & S.T.d.R), CEECUIND/01234/207 (S.T.d.R), and SFRH/BD/137099/2018 (A.C.R.), by an ERC Advanced Investigator award ERC-ADG-2014 671027 attributed to E.H., Sir Henry Wellcome Postdoctoral Fellowship (J.J.Z.), Japan Society for the Promotion of Science KAKENHI grants (JP17KK0143 and JP20K06484 to Y.S., JP19H04970, JP19H03158 and JP20H05393 to K.M., JP17K17719 to T.H., JP18H05534 to H.Ku, JP18H05527 and JP20H00456 to Y.O., JP17H01417 and JP18H05527 to H.Ki), and Japan Science and Technology Agency (JST) CREST JPMJCR16G1 to T.K., H.Ku, Y.O. and H.Ki, PREST JPMJPR2026 to K.M., and ERATO JPMJER1901 to H.Ku. J.J.Z. is supported by core funding of The Novo Nordisk Foundation Center for Stem Cell Biology (Novo Nordisk Foundation grant number NNF17CC0027852). Open Access funding enabled and organized by Projekt DEAL.

## Author contributions

JJZ, EH and HKi involved in conceptualisation and funding acquisition. SJDT, JR, ALS, MH, AO, TH and SB-A involved in methodology. JN involved in software. SJDT, MH and KM validated the study. SJDT, AB, JJZ, TK and SB-A involved in formal analysis. SJDT, JJZ, ACR, MH, YS and AO investigated the study. ALS, JN

and TH involved in resources. AB involved in data curation. JJZ, YS and HKi wrote the original draft. SJDT, AB, STdR and EH wrote, reviewed and edited the manuscript. SJDT, AB, JJZ, STdR, MH and YS involved in visualisation. JJZ, EH, STdR, YS, YO, HKu and HKi supervised the study. JJZ, YS, HKi and EH involved in project administration.

## Conflict of interest

The authors declare that they have no conflict of interest.

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
