## [Review Process File · EMBO Reports]

H4K20me1 and H3K27me3 are concurrently loaded onto the inactive X chromosome but dispensable for inducing gene silencing

Sjoerd Tjalsma, Mayako Hori, Yuko Sato, Aurelie Bousard, Akito Ohi, Ana Raposo, Julia Roensch, Agnes Le Saux, Jumpei Nogami, Kazumitsu Maehara, Tomoya Kujirai, Tetsuya Handa, Sandra Bagés-Arnal, Yasuyuki Ohkawa, Hitoshi Kurumizaka, Simão Teixeira da Rocha, Jan Zylicz, Hiroshi Kimura, and Edith Heard

DOI: [10.15252/embr.202051989](https://doi.org/10.15252/embr.202051989)

Corresponding author(s): Jan Zylicz (jan.zylicz@sund.ku.dk) , Hiroshi Kimura (hkimura@bio.titech.ac.jp), Edith Heard (edith.heard@embl.org)

Review Timeline:

Submission Date:	31st Oct 20
Editorial Decision:	11th Dec 20
Revision Received:	22nd Dec 20
Accepted:	7th Jan 21

Editor: Esther Schnapp

Transaction Report: This manuscript was transferred to EMBO reports following peer review at The EMBO Journal.

We would like to thank the reviewers for their constructive comments and suggestions which have allowed us to significantly improve the manuscript. Below we list our specific responses to the comments.

Referee #1:

This submission utilizes the unique resource of mintbodies to allow tracking of histone modification acquisition in living cells in conjunction with several mouse ESC lines to examine the acquisition of histone modifications during X-chromosome inactivation.

The technical accomplishment of a H3K27me3 mintbody is well-described; and, in combination with the previously developed H4K20me1 mintbody further combined with sgRNA loci to track the X chromosomes or the previously reported Bgl stem loops knocked into Xist allowing in vivo analysis. As H4K20me1 is less well studied, additional experiments included allele-specific ChIP for H4K20me1 for comparison to previous results for H3K27me3, and the previously used DeltaBC and Full-length inducible cell lines were now assessed for H4K20me1. Overall, this is an important series of technical advances and experimental results.

Specific comments:

1. The introduction provides a good literature review, but surprisingly does not seem to include: Methods Mol Biol 2018;1861:91-102. doi: 10.1007/978-1-4939-8766-5_8. Visualizing the Dynamics of Inactive X Chromosomes in Living Cells Using Antibody-Based Fluorescent Probes Yuko Sato, Timothy J Stasevich, Hiroshi Kimura

We thank the reviewer for the comment. We have added this reference to page 3 paragraph 2 line 12.

2. I would like to have seen a quantitation showing levels of Xist after DOX induction. The text and figures left me somewhat confused - the FISH for Xist seemed visible at 3 hours, with H3K27me3/H4K20me1 perhaps 45' later; while the text said ChIP showed enrichment 8 hours after DOX.

We have now included quantification of Xist accumulation using RNA FISH (**Figure EV5B**) and live imaging (**Figure EV4F**). Indeed after 4 hrs of DOX induction Xist is induced in ~48% and ~40% of cells as measured by FISH or live-imaging respectively. From live-imaging we can detect first H3K27me3/H4K20me1 enrichment after 45 min of Xist-cloud appearance. Thus at the 4hrs DOX time point only about ~30% of cells actually show H3K27me3/H4K20me1-enrichment detected by live-imaging. This likely constitutes an insufficient fraction of cells to detect robust enrichment at the whole X chromosome using nChIP-seq at 4hs. Maximum Xist induction efficiency is after about 6hrs of DOX treatment (>60%), thus by the 8hrs time point nearly all of these cells show H3K27me3/H4K20me1 enrichment, which is consistent with our ChIPseq analysis.

3. There are reports of H3K27me3 enriching to sites of 'pre-existing' marks, so in Figure 4 would it be possible to show the '0' time point as well? Panel 4F shows an interesting correlation, but it would also be useful to see each relative enrichment taking into account the proportion of the X covered by the feature (ie where is most of the enrichment seen)? Keeping the scales of the correlation figures the same would

also help comparisons.

We have now included the plot showing H3K27me3 prior to DOX addition (Figure EV5E). This analysis shows that many of the regions accumulating H4K20me1 upon Xist induction show pre-marking of H3K27me3. We have not included all windows on the same scale, since the window size varies significantly between types of feature and we therefore believe that they should not be directly cross-compared.

4. In Figure 5 the blue and red are challenging to distinguish and black was not labelled. Could the charts be separate? It seems that the H4K20me1 signal is stronger, yet it has less read enrichment - some discussion to clarify would be helpful.

We decided to visualise data in a way that is consistent with previous work (Zylicz et al, 2019). Since we are looking at ratios between B6/Cast, when separating reads it becomes much more difficult to interpret than an overlay. However to further clarify the figure we now added a legend for the black color which is the overlap between B6 and Cast reads. As for the varying levels of enrichment, it is important to note that both marks behave rather differently within these regions. We are looking at two initially active genes not pre-marked by H3K27me3 (hence lower signal at t=0hrs) but with strong enrichment of H4K20me1 in the gene body (hence improved signal at t=0hrs). This point is discussed on page 7 paragraph 2: "While investigating genes that are initially active and then become silenced following *Xist* induction, we observed that H4K20me1 is strongly biallelically enriched (pre-marked) at transcribed gene bodies, prior to silencing (**Fig. 5A**). This contrasts strikingly with the distribution of H3K27me3, which never pre-marks initially active genes (Zylicz et al., 2019).

5. For the experiments in the male line with full-length of the BC deletion - is the DOX induction still 24 hours? Was any lethality observed? The statement is made "the global level of H4K20me1 was comparable", but further discussion is warranted. Was this by western analysis? Cell heterogeneity is commented upon, but presumably assessment of a large cell population would compensate for cell differences?

The experiment was performed 48hrs upon DOX treatment and in both samples some cell death was detected. We now clarified this in the legend of Fig 3. D "Cells induced for 48hrs in the absence of LIF were used for combined IF and RNA FISH."

Cell death has been previously reported (Bousard *et al*, 2019). We have compared the levels by IF. Most of the H4K20me1 is not located at the Xi but at mitotic chromosomes and such it would be impossible to detect any Xi-specific changes using Western Blotting.

6. It seems that assessing H4K20me1 when PRC1/PRC2 (or HNRNPK) were knocked out or inhibited might address whether H4K20me1 is downstream of those pathways, as suggested by: Schoeftner, S., Sengupta, A. K., Kubicek, S., Mechtler, K., Spahn, L., Koseki, H., et al. (2006). Recruitment of PRC1 function at the initiation of X inactivation independent of PRC2 and silencing. *The EMBO Journal*, 25(13), 3110-3122. <http://doi.org/10.1038/sj.emboj.7601187> which suggested that H3K27me3 was involved in the establishment of H4K20me1 on the inactive X by Xist.

We thank the reviewer for this valuable comment. Indeed Schoeftner et al., has revealed that there is partial loss of H4K20me1 enrichment at the Xi in Eed mutant cells. This indicates that H3K27me3 plays some role in allowing for H4K20me1

enrichment (e.g. via L3MBTL1) but is only part of the mechanism. Other pathways most likely occur and these indeed could depend on PRC1/HNRNPK. Because of time constraints we are unable to provide definitive mechanistic data for this. Instead we have extended and corrected our discussion. p9 paragraph 2 now includes: “In line with this model, loss of EED, a core PRC2 component, results in partially reduced H4K20me1 enrichment at the Xi (Schoeftner *et al*, 2006). Thus PRC2 contributes to H4K20me1 accumulation but other pathways must also operate.”

A typographical error: time laps was used instead of time lapse.

We have now corrected this error.

Referee #2:

The authors studied spatio-temporal dynamics of histone modifications, H3K27me3 and H4K20me1, on the inactive X chromosomes in mouse ES cells using mintbodies against these histone modifications in combination with Blg-tagged Xist RNA visualized by BlgG-FP. They demonstrated that these modifications started to be enriched on the X chromosome about 45 min after the accumulation of Xist RNA became evident. Both H3K27me3 and H4K20me1 initially accumulated in similar dynamics, but while H3K27me3 continued to accumulate over time, the enrichment of H4K20me1 reached a plateau at around 2.5 hours after Xist accumulation. They subsequently took advantage of hybrid female ESCs carrying an inducible Xist allele and carried out allele-specific nChIP-seq of H4K20me1 at several time point after induction of Xist by dox. Comparison of the results with those of H3K27me3 they previously reported confirmed the live-imaging analysis. It also revealed that although both marks accumulated in a similar fashion in the intergenic regions and inactive gene bodies, their distributions differed in the initially active gene bodies, where H4K20me1 was biallelically enriched prior to XCI and stayed unchanged upon induction of Xist. Finally, the authors explored the relationship of H4K20me1 enrichment and deposition of H3K27me3 during XCI using male ESCs carrying an inducible Xist allele deleted for the B and C repeats and found that H4K20me1 was not enriched on the X chromosome coated with Xist RNA lacking these repeats. This suggested that H4K20me1 enrichment depended on the same regions of Xist RNA involved in the recruitment of PRC1 and PRC2.

This is a first detailed description of the kinetics and distribution of H4K20me1 on the X chromosome during XCI. It contains some potentially interesting findings to think about the role of H4K20me1 in XCI. A novelty of this study is simultaneous live-imaging of the histone modifications and Xist RNA during XCI. The finding that while the enrichment of H4K20me1 follows the pattern of H3K27me3 in the intergenic regions and the inactive gene loci on the X chromosome during XCI, gene bodies of those undergoing inactivation upon Xist upregulation are enriched with H4K20me1 on both Xs and stay constant even upon induction of Xist is unexpected. Experiments were carefully performed and the results were reasonably documented. It is, however, questionable whether the study provides significant advances in our understanding of the role of H4K20me1 in the process of XCI. Nonetheless, I think that the authors might want to address the following issues to improve the manuscript regardless of whether or not the manuscript will be sent to another round of review for further consideration.

Major points

What proportion of cells exhibited the Xist cloud at each time point following induction of Xist by dox in the experiment of Figure 4? I am wondering if an increase in the enrichment of these modifications (Figure 4 B and C) represents an increase in the respective modifications on the X coated with Xist RNA or an increase in the population of cells harboring the Xist cloud.

We have now included quantification of Xist induction in Fig. EV5B. Indeed there is some increase in the fraction of cells showing Xist clouds between 4-8hrs. However, after 8hrs the number of cells with Xist enrichment is stable. Therefore the later increase in enrichment is due to histone mark accumulation and not Xist induction.

They found that although an enrichment of H3K27me3 increased over time, an enrichment of H4K20me1 reached a plateau. This may reflect further methylation of H4K20me1 to H4K20me2/3. Although no enrichment of H4K20me3 has not been observed on the inactive X chromosome in the mouse by immune fluorescence staining, nChIP-seq may reveal an increase in H4K20me2/3 levels in some regions on the Xi. In fact, the authors pointed out an interesting possibility that an enrichment of H4K20me1 might not have facilitated by SETD8-mediated monomethylaiton but by demethylation of H4K20me2/3. Pattern of H4K20me1 distribution on Xi may be created by demthylation. In this connection, it is of interest to explore changes in the distribution of H4K20me2/3 on the X chromosomes during XCI.

We thank the reviewer for this comment. We have now included IF analysis revealing that only H4K20me1 and not H4K20me2 or H4K20me3 accumulates at the Xi (**Fig EV4A**). In light of this finding we decided not to perform ChIP-seq for H4K20me2 due to high-costs of such experiments and the fact that they are unlikely to provide valuable functional insights. Since H4K20me2/3 demethylation might contribute to increased H4K20me1 levels we have extensively discussed this possibility at page 9 paragraph 2.

They showed that the B and C repeats of Xist RNA were required for the enrichment of H4K20me1. It is possible that the enrichment of H4K20me1 requires H2AK119ub as is the case of initial recruitment of H3K27me3 or depends on H3K27me3 thus recruited by H2K119ub. Or, it is also possible that the machinery for H4K20me1 requires the B and C repeats on Xi-loading in a manner independent of PRC1 and PRC2. It is of interest to see the enrichment of H4K20me1 in cells deficient for either both PCGF3 and 5 or EED.

We thank the reviewer for this comment. Indeed the increase of H4K20me1 at the Xi can relate to either PRC1/HNRNPK recruitment or that of PRC2. We have now included discussion of the Schoefnter et al. 2006 publication which revealed decreased H4K20me1 enrichment at Xi in Eed-KO cells. This indicates that PRC2 contributes to H4K20me1 enrichment but other pathways are also at play. We thus expanded the discussion of the possible function of PRC1/HNRNPK in this process. Unfortunately due to time constraints we are unable to provide precise mechanistic data. Instead we have extended and corrected our discussion. p9 paragraph 2 now includes: "In line with this model, loss of EED, a core PRC2 component, results in partially reduced H4K20me1 enrichment at the Xi (Schoeftner *et al.*, 2006). Thus PRC2 contributes to H4K20me1 accumulation but other pathways must also operate."

Minor points

S10ph in a Table in Fig. EV2B should be S28ph.

This is now corrected.

Referee #3:

The manuscript entitled "Tracking H3K27me3 and H4K20me1 dynamics during XCI reveals similarities in recruitment mechanism" by Tjalsma et al. uses histone modification specific mintbodies that can be used within living cells for studying the process of X chromosome inactivation in mouse embryonic stem cells. The authors report on the successful design of a new mintbody construct that can recognize H3K27me3 and used to fluorescently label chromatin enriched in this modification within living cells. They use this novel tool together with a previously described mintbody for H4K20me1 and cytological staining techniques to characterize the timing of chromatin modifications following Xist expression. This is a technical advanced system that shows the enrichment of both histone marks with some delay after Xist accumulation over the inactive X chromosome. Further allele specific ChIP is used to characterize the distribution of H4K20me1 and compare it to the pattern of H3K27me3. Although, the data is technically of high quality it is difficult to make solid mechanistic conclusions. Indeed, the authors state that a similar mechanism and timing might be engaged in recruiting H3K27me3 and H4K20me1, but the patterns observed by ChIP do only overlap in not necessarily the most specific intergenic regions. There are substantial differences on the X and autosomes. This suggests that no clear call could be made for an overlapping mechanism of recruitment. The alleged similarity in timing is also not easy to capitalize on as there will be undoubtedly threshold effects when the mintbodies give clear signals. An experiment shows that H4K20me1 enrichment is dependent on Xist repeat B and C similar to what has been reported before for H3K27me3 is noteworthy. However, the lack of an interaction of Set7 with hnRNPK would possibly suggest that it is downstream of chromatin modifications of the Polycomb system similar to SmcHD1. This would suggest a different mechanism and likely timing that would not be resolved in the experimental setup. This scenario is not ruled out by the present version. A solid conclusion would be that the histone marks become visible after Xist becomes detectable but this is largely expected. Therefore the mechanistic advance in the present version is not well supported (also the title) by the experimental data. Irrespective the manuscript does have technical merits.

Specific points

1. Page 6 from which data do the authors arrive at "This suggests that the two marks may share a similar recruitment mechanism to the Xi, even though their dynamics are not identical". The microscopy observation of enrichment does not say anything about a mechanism. The conclusion is not supported by the data.

We agree with the reviewer that we have not provided conclusive mechanistic data uncovering the deposition mechanism of H4K20me1 at the Xi. Therefore we have modified the text which now reads: "This shows that the two marks share similar dynamics of enrichment but only at the early stages of XCI."

2. Page 6 "...although ... And H4K20me1 are both initially enriched soon after Xist RNA...": The enrichment of H4K20me1 is much weaker than that of H3K27me3 and could be a result of compacting or higher density of chromatin of the Xi. A picture of metaphase chromosomes should be included showing the enrichment over autosomes at the same compactions state can be demonstrated with the mintbody.

During live imaging Xist signal is lost thus we cannot track the enrichment signal from

H4K20me1. What is more, during mitosis the global levels of H4K20me1 are drastically increased at the autosomes (Oda *et al*, 2009). In our data we could not detect H4K20me1 enrichment at the presumptive Xi, during mitosis at early differentiation stages [Figures for referees not shown.] . However, at later stages of XCI there was apparent enrichment of H4K20me1 at one chromosome even during mitosis (see (Sato *et al*, 2016) Figure 4B). Thus it seems that H4K20me1 enrichment at the early stages of Xi precedes global accumulation of H4K20me1 during mitosis. What is more, it might well be that H4K20me1 enrichment is linked to chromatin compaction, however it is not due to an imaging artefact. Indeed our ChIP-seq analysis confirmed H4K20me1 accumulation at the Xi but not at the active X chromosome. Thus two orthogonal methods confirmed H4K20me1 enrichment at the Xi.

3. Page 6: "... we normalized both marks to their effective dynamic range ..." is not clear. The marks are visualized by different fluorophores. Comparison should be made with both mintbodies linked to GFP to ensure the same sensitivity of the fluorescent marker is compared.

The fluorophore-dependent sensitivity is not an issue in this case, because both show global nuclear background which we use for normalisation during the measurement of mintbody enrichment on Xist-positive regions.

4. The authors refer to recruitment mechanism for H4K20me1 throughout page 6. This is a presumption as it has not been shown that Set7 is recruited or interacts with Xist. Without data showing that the histone methylase is equally enriched on Xi a different mechanism cannot be ruled out. The conversion of H4K20me1 to me2 and Me3 could be inhibited and therefore the enrichment would be caused by an indirect effect preventing di and tri methylation of K20. To rule this out the level of H4K20me3 and me2 needs to be determined after Xist expression and shown that it is unaffected. A similar prevention of me2 and me3 gives rise to a perceived "enrichment" of H3K27me1 at pericentric chromatin, where PRC2 is blocked by H3K9me3. This type of mechanism would also be consistent with the relatively low levels of enrichment observed for H4K20me1.

This is an important point raised by the reviewer and it is now extensively discussed in the manuscript. Indeed, it remains unclear if it is the enrichment of PrSet7 or rather prevention of further methylation at H4K20. We now report that neither H4K20me2 or me3 are enriched along the Xi (**Fig. EV4A**). This however does not exclude the possibility that SUV420H1/2 shows reduced activity at the Xi thus leading to

accumulation of H4K20me1. We were unable to directly address this point in this manuscript however we have now discussed this on page 9 paragraph 2: “Finally, it is also possible that H4K20me1 enrichment at the Xi is due to hindered conversion to higher methylation states by SUV4-20H1/2.”

5. Conceptually, there is a problem in timing. The authors state that PRSet7 activity is maximal in G2/M phase and mitotic cells are excluded from H4K20me1 analysis due to the high levels. I would think this is misleading. In addition, Xist is not on chromatin from early prophase of mitosis. This would provide a restricted window for Xist dependent recruitment of a H4K20me1 methylase. This could mean that a considerable part of the H4K20me1 is established when Xist is not on the chromosome.

As reviewer rightly points out in the previous comment it is possible that H4K20me1 enrichment at the Xi is not due to PrSet7 recruitment but results from reduced conversion to H4K20me2/3 or increased demethylation from H4K20me2/3. The latter two mechanisms are fully compatible with our observation of H4K20me1 enrichment beyond mitosis. Indeed we confirmed this both in LIF-withdrawal and Xist-inducible systems. Finally, despite the cycling nature of PrSet7 it is not totally degraded during the S-phase and there are significant levels of PrSet7 at G2 and early G1 (Oda et al., 2009). Thus it remains plausible that H4K20me1 is deposited to Xi at these stages. If PRSet7 is recruited to Xist-rich regions its high local concentration can increase the local H4K20me1 level.

6. Page 7 “..Implying that both marks differ in their correlation with transcription” can you be more specific? How would this be possible when earlier a common recruitment mechanism is proposed?

This point is further explored in the next paragraph:

“While investigating genes that are initially active and then become silenced following Xist induction, we observed that H4K20me1 is strongly biallelically enriched (pre-marked) at transcribed gene bodies, prior to silencing (**Fig. 5A**). This contrasts strikingly with the distribution of H3K27me3, which never pre-marks initially active genes (Zylicz *et al*, 2019). We confirmed that this H4K20me1 enrichment at transcribed genes is a general feature both on the X chromosome (**Fig. 5B**) and autosomes (**Fig. EV5D**). This is in line with previous reports indicating that H4K20me1 correlates with transcriptional elongation and is enriched at active gene bodies (Beck *et al*, 2012; Kapoor-Vazirani & Vertino, 2014; Veloso *et al*, 2014).”

In this publication we point out to similarities in enrichment mechanisms of H3K27me3/H4K20me1, which rely on the BC-region of Xist. However, we do not identify a common recruitment mechanism of PRC2 and PrSet7. We have now modified the title and manuscript text to precisely express this. Finally, we have uncovered that H4K20me1 enrichment at the Xi relies on Xist RNA. On the other hand, at other regions H4K20me1 correlates positively with transcription. This is the key difference with H3K27me3. Thus differing correlation with transcription of both marks is explained by Xist-independent enrichment mechanisms.

7. H2AK119Ub has been suggested to be the primary histone modification for Polycomb recruitment to the Xi. This view is consistent with earlier studies showing loss of Polycomb marks including H3K27me3 in cells with mutations in Pcgf3 and 5.

Therefore one would expect H2Aub be recruited earlier at Xi than H3K27me3. Can this timing be resolved by the ChIP? If not the question would arise how can timing differences large enough be limiting for mechanistic discovery. If the difference cannot be measured would not everything co-occur temporally and therefore "suggest a similar recruitment mechanism" which would be wrong from what is established for Polycomb modifications. This seems to be a principle issue with the analysis in this study.

The reviewer is right to point this out. In fact our analysis from Zylicz et al., 2019 clearly shows that H2AK119Ub becomes enriched at the Xi significantly prior to H3K27me3. We detected very significant enrichment of H2AK119Ub already after 4hrs of DOX treatment and the ED50 value of ~6hrs, for H3K27me3 this is ~14hrs and first enrichment only after 8hrs. For H4K20me1 there is very little enrichment at the 4hrs timepoint and the ED50 value is ~10hrs. Below we plotted dynamics of all three marks, please note rapid H2AK119Ub accumulation.[Figures for referees not shown.]

8. Page 7 "In this sense H4K20me1 seems to behave like an active chromatin mark" is a bit misleading as it seems not to be associated with transcription starts. The enrichment in the gene body is interesting but also K36me3 would likely be seen as repressive to transcription. The association with the inactive X would on balance suggest that no general association with transcription status can be made. I would rather see it repressive if at all.

We have now removed this sentence.

9. Page 9 "Which factors directly target H4K20me1 to the Xi still remains unclear" is an overstatement. The authors have not shown that H4K20me1 is directly targeted to the Xi. It could be there before Xist expression and become apparently enriched by blocking of conversion to H4K20me2 and me3. It would be more interesting to assess if "enrichment" for H4K20me1 would be dependent on H3K27me3 in a Ezh2 mutant background. In this experiment H2Aub could be used as a control for showing that at least enrichment for one chromatin mark is unaffected. A PRC2 mutation would also allow to demonstrate the specificity of the H2K27me3 Mintbody.

We have now rephrased this sentence and it reads: "Which factors directly allow for H4K20me1 enrichment at the Xi still remains unclear however.". As for the mechanism of enrichment, published data indicated that Eed mutation results in only partial depletion of H4K20me1 at the Xi (Schoeftner *et al.*, 2006). This result is now adequately discussed. Thus, other mechanisms must contribute and could involve PRC1/hnRNPK. Finally we have extensively validated mintbody specificity using KDM overexpression and dot-blot analysis (Figure EV2).

10. In Figure 3B a number of bright staining foci become visible from 3 hours. What are these foci? The foci overlapping Xist appear initially less well discernible and only become brighter at 5 hours. Could this be pericentric regions and indicate some crossreactivity of the Mintbody with H4K20me2 or H4K27me3? A co-staining with H3K9me3 should be shown to rule out enrichment over pericentric regions.

In our experiments we did not see an increase in H3K27me3 at foci, rather there is an increase of global signal indicating very low levels of bleaching (see Movie 2). From our experience foci of H3K27me3 are typical of ESCs grown under 2iLif conditions. Depending on the focal plane and cell cycle, some bright foci other than Xi were also found in Fig. 2, which is one of the reasons why the Xist inducible system is more suitable for analyzing the accumulation kinetics. Exclusion of H4K20me1 and H3K27me3 Mintbodies from pericentric regions were shown previously (Sato et al, 2016) and Fig. 1, respectively.

Minor points

a) Page numbers are missing, please include.

This was now included.

REFERENCES

- Beck DB, Oda H, Shen SS, Reinberg D (2012) PR-Set7 and H4K20me1: at the crossroads of genome integrity, cell cycle, chromosome condensation, and transcription. *Genes Dev* 26: 325-337
- Bousard A, Raposo AC, Żylicz JJ, Picard C, Pires VB, Qi Y, Gil C, Syx L, Chang HY, Heard E *et al* (2019) The role of Xist-mediated Polycomb recruitment in the initiation of X-chromosome inactivation. *EMBO reports*
- Kapoor-Vazirani P, Vertino PM (2014) A dual role for the histone methyltransferase PR-SET7/SETD8 and histone H4 lysine 20 monomethylation in the local regulation of RNA polymerase II pausing. *J Biol Chem* 289: 7425-7437
- Oda H, Okamoto I, Murphy N, Chu J, Price SM, Shen MM, Torres-Padilla ME, Heard E, Reinberg D (2009) Monomethylation of histone H4-lysine 20 is involved in chromosome structure and stability and is essential for mouse development. *Mol Cell Biol* 29: 2278-2295
- Sato Y, Kujirai T, Arai R, Asakawa H, Ohtsuki C, Horikoshi N, Yamagata K, Ueda J, Nagase T, Haraguchi T *et al* (2016) A Genetically Encoded Probe for Live-Cell Imaging of H4K20 Monomethylation. *J Mol Biol* 428: 3885-3902
- Schoeftner S, Sengupta AK, Kubicek S, Mechtler K, Spahn L, Koseki H, Jenuwein T, Wutz A (2006) Recruitment of PRC1 function at the initiation of X inactivation independent of PRC2 and silencing. *EMBO J* 25: 3110-3122
- Veloso A, Kirkconnell KS, Magnuson B, Biewen B, Paulsen MT, Wilson TE, Ljungman M (2014) Rate of elongation by RNA polymerase II is associated with specific gene features and epigenetic modifications. *Genome Res* 24: 896-905
- Zylicz JJ, Bousard A, Zumer K, Dossin F, Mohammad E, da Rocha ST, Schwalb B, Syx L, Dingli F, Loew D *et al* (2019) The Implication of Early Chromatin Changes in X Chromosome Inactivation. *Cell* 176: 182-197 e123

Dear Jan,

Thank you for the submission of your revised manuscript and the pleasant video chat regarding potential revisions. I have now re-discussed your manuscript with my colleagues here, and I am happy to say that we have decided that we can offer to publish it, if all novel aspects of your study and their relevance for the broader chromatin community can be more clearly presented.

Please also address the referee comments below in a point-by-point response, as the comments will be part of our transparent peer-review process file (RPF).

A few editorial changes are also required:

- Your manuscript has 5 main figures, but is layed out as a full length article. Please either combine the results and discussion sections to publish your paper as a short report with a maximum of 5 main figures, or add one main figure to the manuscript file to publish it as an article. The short report should not have more than 27.000 characters including spaces but excluding references and materials and methods. You can find more information in our guide to authors online.
- Please list up to 5 keywords in your manuscript file.
- Please add the Data Availability Section (DAS) to the end of the materials and methods. Please add accession numbers and websites that directly link to your deposited data to the DAS. The data need to be freely accessible from the day of online publication of your manuscript.
- Please rename the supplementary table "Table EV1", and remove the table legend from the manuscript and add it to the first tab of the excel file.
- Please also remove the movie legends from the manuscript and zip them with each respective movie file. One zipped file per movie needs to be uploaded.
- I attach to this email a related manuscript file with comments by our data editors. Please address all comments in the final manuscript file.
- As we discussed, please modify the abstract and perhaps the title to better reflect the advances provided by your study. Please describe the new findings in the abstract in present tense.

EMBO press papers are accompanied online by A) a short (1-2 sentences) summary of the findings and their significance, B) 2-3 bullet points highlighting key results and C) a synopsis image that is exactly 550 pixels wide and 200-600 pixels high (the height is variable). You can either show a model or key data in the synopsis image. Please note that text needs to be readable at the final size. Please send us this information along with the revised manuscript.

I look forward to seeing a final version of your manuscript as soon as possible.
Please let me know if you have any questions or comments.

Referee #1:

The authors have addressed the issues raised by reviewer 3 and myself by essentially editing the text. Although I think their responses on both reviewers are largely acceptable, I would like to comment a couple of things.

Intergenic regions and transcriptionally repressed X-linked genes in undifferentiated ESCs acquire H4K20me1 on the X chromosome upon induction of Xist in cis in a manner similar to H3K27me3, whereas gene body of transcribed genes on both Xs are enriched with H4K20me1 from the beginning and this state does not change for 24 hours after induction of Xist. It is not clear if SETD8 mediates deposition of H4K20me1 or demethylases mediate demethylation from H4K20me2/3 to H4K20me1 or both take place to create an observed distribution of H4K20me1 on the inactive X. I think that knowing the dynamics of H4K20 methylation state on the X undergoing inactivation would provide an insight into our understanding of how the distribution of H4K20me1 is created on the inactive X. Reviewer 3 also pointed out the importance of studying the level of H4K20me2/3 after Xist expression (major comment 4). Although I do not insist they do ChIP-seq for H4K20me2/3 for this descriptive study, I am sorry to hear that the authors felt that further analysis for the distribution of H4K20me2/3 would not provide valuable functional insights.

Since the authors are reporting the similarities in enrichment dynamics of H3K27me3 and H4K20me1, I wanted to know in this study if the enrichment of H4K20me1 also depended on H2AK119ub. They imply this possibility by discussing the presence of a mechanism other than the one dependent on H3K27me3 for the enrichment of H4K20me1 on the inactive X, based on a previous study showing that the enrichment of H4K20me1 was only slightly diminished in EED mutant ESCs. In response to reviewer 3's comment 7, the authors showed dynamics of H2AK119ub, H3K27me3, and H4K20me1 enrichment. Although this revealed accumulation of H2AK119ub on the X preceded that of H4K20me1 upon Xist induction in cis, it remains unanswered at all if H2AK119ub is required for the accumulation of H4K20me1. I am just wondering if the authors think it reasonable to leave this experiment behind for the future study due to the time constraints although all reviewers raised this point in the previous round of review.

We are grateful for the referees comments and interest in our manuscript. Below is our point-by-point response to referee's comments :

Referee #1:

The authors have addressed the issues raised by reviewer 3 and myself by essentially editing the text. Although I think their responses on both reviewers are largely acceptable, I would like to comment a couple of things.

Intergenic regions and transcriptionally repressed X-linked genes in undifferentiated ESCs acquire H4K20me1 on the X chromosome upon induction of Xist in cis in a manner similar to H3K27me3, whereas gene body of transcribed genes on both Xs are enriched with H4K20me1 from the beginning and this state does not change for 24 hours after induction of Xist. It is not clear if SETD8 mediates deposition of H4K20me1 or demethylases mediate demethylation from H4K20me2/3 to H4K20me1 or both take place to create an observed distribution of H4K20me1 on the inactive X. I think that knowing the dynamics of H4K20 methylation state on the X undergoing inactivation would provide an insight into our understanding of how the distribution of H4K20me1 is created on the inactive X. Reviewer 3 also pointed out the importance of studying the level of H4K20me2/3 after Xist expression (major comment 4). Although I do not insist they do ChIP-seq for H4K20me2/3 for this descriptive study, I am sorry to hear that the authors felt that further analysis for the distribution of H4K20me2/3 would not provide valuable functional insights.

We agree with the referee that H4K20me1 accumulation during XCI might be a result of inhibited conversion towards higher methylation states. Because of this possibility we have tracked the levels of H4K20me2 and H4K20me3 at the inactive X by immunofluorescence. We have not found a marked depletion of H4K20me2 at the inactive X indicating that reduced SUV420H1/2 activity is unlikely. While we do not preclude this mechanism, in our view it remains improbable.

Since the authors are reporting the similarities in enrichment dynamics of H3K27me3 and H4K20me1, I wanted to know in this study if the enrichment of H4K20me1 also depended on H2AK119ub. They imply this possibility by discussing the presence of a mechanism other than the one dependent on H3K27me3 for the enrichment of H4K20me1 on the inactive X, based on a previous study showing that the enrichment of H4K20me1 was only slightly diminished in EED mutant ESCs. In response to reviewer 3's comment 7, the authors showed dynamics of H2AK119ub, H3K27me3, and H4K20me1 enrichment. Although this revealed accumulation of H2AK119ub on the X preceded that of H4K20me1 upon Xist induction in cis, it remains unanswered at all if H2AK119ub is required for the accumulation of H4K20me1. I am just wondering if the authors think it reasonable to leave this experiment behind for the future study due to the time constraints although all reviewers raised this point in the previous round of review.

We agree with the reviewers comment that the involvement of H2AK119Ub in targeting H4K20me1 accumulation remains unknown. Indeed, the experiments proposed by the reviewer would be of interest. In order to rigorously address the function of H2AK119Ub one would have to construct new inducible double-knockout Ring1A/B ESCs in our TX1072 background. The interpretation of such a laborious experiment would be complicated by the fact that we would dramatically affect the chromatin state of the X chromosome even before it becomes coated by Xist RNA. Our previous data indicated that pre-marking of genes by H2K119Ub might well have an important role in XCI and maybe even Xist RNA coating (Zylicz *et al*, 2019). In light of these complications we feel that the proposed experiments are outside the scope of this manuscript.

Zylicz JJ, Bousard A, Zumer K, Dossin F, Mohammad E, da Rocha ST, Schwalb B, Syx L, Dingli F, Loew D *et al* (2019) The Implication of Early Chromatin Changes in X Chromosome Inactivation. *Cell* 176: 182-197 e123

Jan Zylicz
The Novo Nordisk Foundation Center for Stem Cell Biology
Copenhagen
Denmark

Dear Jan,

Thank you for the submission of your revised manuscript. It looks all good now, except that I would add the word "chromosome" to the title so that it reads:

"H4K20me1 and H3K27me3 are concurrently loaded onto the inactive X chromosome but dispensable for inducing gene silencing"

Please let me know if this title is OK with you.

I am very pleased to accept your manuscript for publication in the next available issue of EMBO reports. Thank you for your contribution to our journal.

Please note that under the DEAL agreement of German scientific institutions with our publisher Wiley, you could be eligible for publication of your article in the open access format in a way that is free of charge for the authors, given that one of the corresponding authors of your manuscript is in Germany. Please contact either the administration at your institution or our publishers at Wiley (emboreports@wiley.com) for further questions.

At the end of this email I include important information about how to proceed. Please ensure that you take the time to read the information and complete and return the necessary forms to allow us to publish your manuscript as quickly as possible.

As part of the EMBO publication's Transparent Editorial Process, EMBO reports publishes online a Review Process File to accompany accepted manuscripts. As you are aware, this File will be published in conjunction with your paper and will include the referee reports, your point-by-point response and all pertinent correspondence relating to the manuscript.

If you do NOT want this File to be published, please inform the editorial office within 2 days, if you have not done so already, otherwise the File will be published by default [contact: emboreports@embo.org]. If you do opt out, the Review Process File link will point to the following statement: "No Review Process File is available with this article, as the authors have chosen not to make the review process public in this case."

Should you be planning a Press Release on your article, please get in contact with emboreports@wiley.com as early as possible, in order to coordinate publication and release dates.

Thank you again for your contribution to EMBO reports and congratulations on a successful publication. Please consider us again in the future for your most exciting work.

Best wishes, and happy new year !

Esther

THINGS TO DO NOW:

You will receive proofs by e-mail approximately 2-3 weeks after all relevant files have been sent to our Production Office; you should return your corrections within 2 days of receiving the proofs.

Please inform us if there is likely to be any difficulty in reaching you at the above address at that time. Failure to meet our deadlines may result in a delay of publication, or publication without your corrections.

All further communications concerning your paper should quote reference number EMBOR-2020-51989V2 and be addressed to emboreports@wiley.com.

Should you be planning a Press Release on your article, please get in contact with emboreports@wiley.com as early as possible, in order to coordinate publication and release dates.

Corresponding Author Name: Jan Zyllicz

Manuscript Number: EMBOR-2020-51989-T